# Intermittent Hypoxia Induces Cognitive Dysfunction and Hippocampal Gene Expression Changes in a Mouse Model of Obstructive Sleep Apnea

**DOI:** 10.3390/ijms26157495

**Published:** 2025-08-03

**Authors:** Kenta Miyo, Yuki Uchida, Ryota Nakano, Shotaro Kamijo, Masahiro Hosonuma, Yoshitaka Yamazaki, Hikaru Isobe, Fumihiro Ishikawa, Hiroshi Onimaru, Akira Yoshikawa, Shin-Ichi Sakakibara, Tatsunori Oguchi, Takuya Yokoe, Masahiko Izumizaki

**Affiliations:** 1Department of Physiology, Showa Medical University School of Medicine, Tokyo 142-8555, Japan; miyoken1993@med.showa-u.ac.jp (K.M.); oni@med.showa-u.ac.jp (H.O.); masahiko@med.showa-u.ac.jp (M.I.); 2Department of Respiratory Medicine, Showa Medical University Fujigaoka Hospital, Yokohama 227-8501, Japan; takknyo@med.showa-u.ac.jp; 3Department of Physiology, Showa Medical University Graduate School of Pharmacy, Tokyo 142-8555, Japan; nakano.r@cmed.showa-u.ac.jp (R.N.); kamijo.s@med.showa-u.ac.jp (S.K.); 4Division of Medical Pharmacology, Department of Pharmacology, Showa Medical University School of Medicine, Tokyo 142-8555, Japan; masa-hero@med.showa-u.ac.jp (M.H.); em21h011@edu.showa-u.ac.jp (H.I.); t.oguchi@med.showa-u.ac.jp (T.O.); 5Division of Toxicology, Showa University Graduate School of Pharmacy, Tokyo 142-8555, Japan; yamazaki.y@pharm.showa-u.ac.jp; 6Center for Biotechnology, Showa Medical University, Tokyo 142-8555, Japan; f-ishikawa@pharm.showa-u.ac.jp; 7Division of Health Science Education, Showa Medical University School of Nursing and Rehabilitation Sciences, Yokohama 226-8555, Japan; yoshi-aki@nr.showa-u.ac.jp; 8Laboratory for Molecular Neurobiology, Faculty of Human Sciences, Waseda University, Tokorozawa 359-1192, Japan; sakakiba@waseda.jp

**Keywords:** obstructive sleep apnea syndrome, Intermittent hypoxia, cognitive dysfunction, hippocampus, gene expression

## Abstract

Obstructive sleep apnea syndrome (OSAS) is characterized by cycles of decreased blood oxygen saturation followed by reoxygenation due to transient apnea. Cognitive dysfunction is a complication of OSAS, but its mechanisms remain unclear. Eight-week-old C57BL/6J mice were exposed to intermittent hypoxia (IH) to model OSAS, and cognitive function and hippocampal gene expression were analyzed. Three groups were maintained for 28 days: an IH group (oxygen alternating between 10 and 21% in 2 min cycles, 8 h/day), sustained hypoxia group (SH) (10% oxygen, 8 h/day), and control group (21% oxygen). Behavioral tests and RNA sequencing (RNA-seq) analysis were performed. While Y-maze test results showed no differences, the IH group demonstrated impaired memory and learning in passive avoidance tests compared to control and SH groups. RNA-seq revealed coordinated suppression of mitochondrial function genes and oxidative stress response pathways, specifically in the IH group. RT-qPCR showed decreased *Lars2*, *Hmcn1*, and *Vstm2l* expression in the IH group. Pathway analysis showed the suppression of the KEAP1-NFE2L2 antioxidant pathway in the IH group vs. the SH group. Our findings demonstrate that IH induces cognitive dysfunction through suppression of the KEAP1-NFE2L2 antioxidant pathway and downregulation of mitochondrial genes (*Lars2*, *Vstm2l*), leading to oxidative stress and mitochondrial dysfunction. These findings advance our understanding of the molecular basis underlying OSAS-related cognitive impairment.

## 1. Introduction

Obstructive sleep apnea syndrome (OSAS) is the most common sleep-related breathing disorder, with a prevalence of 13–33% in men and 6–19% in women [1], affecting approximately 1 billion people worldwide [2], with this prevalence continuing to rise due to increasing obesity rates and population aging [3]. The fundamental pathophysiological feature of OSAS is recurrent upper airway obstruction during sleep, leading to cycles of hypoxia followed by reoxygenation [4].

OSAS presents diverse clinical manifestations, including snoring, excessive daytime sleepiness, and choking sensations during sleep. OSAS significantly affects multiple organ systems and increases the risk of hypertension [5], type 2 diabetes [6], and cardiovascular diseases [7]. Among these complications, cognitive dysfunction has emerged as an important complication of OSAS [8]. Patients with OSAS experience impaired attention, memory deficits, learning difficulties, and executive dysfunction [9,10]. Moreover, OSAS has been associated with increased risk of mild cognitive impairment and dementia [11].

The hippocampus plays a central role in memory formation and consolidation. Neuroimaging studies have revealed hippocampal atrophy in OSAS patients, correlating with decreased memory and learning functions [12]. These structural and functional changes represent a significant clinical burden, as cognitive dysfunction substantially impacts patients’ daily functioning and quality of life. At the cellular level, OSAS impairs synaptic plasticity in the hippocampal CA1 region and reduces neuronal numbers in both CA1 and CA3 regions [13,14]. However, the precise molecular mechanisms underlying OSAS-induced cognitive dysfunction remain poorly understood.

Intermittent hypoxia (IH) is a core pathophysiological condition in OSAS. In mouse IH models, cycling oxygen concentrations are used to reproduce hypoxia-reoxygenation cycles that mimic OSAS. Mice exposed to IH have shown impairments in spatial cognition, working memory, and learning in behavioral tests such as the Morris water maze test [15], Y-maze test [16], and passive avoidance test [17]. These cognitive dysfunctions involve not only decreased oxygen utilization due to hypoxia but also oxidative stress resulting from increased production of reactive oxygen species (ROS) during reoxygenation [18,19,20].

While IH-induced cognitive dysfunction has been demonstrated, studies that distinguish and evaluate the effects of hypoxia and reoxygenation in IH are limited, and the precise roles of specific genes and molecular pathways that cause cognitive dysfunction in the hippocampus remain unclear. Previous studies have established that repeated hypoxia-reoxygenation cycles increase ROS production [18], with sustained hypoxia (SH) typically leading to adaptive responses while intermittent hypoxia generates repeated oxidative stress that may overwhelm cellular antioxidant defenses [21,22,23,24].

Our working hypothesis was that cyclical reoxygenation in IH, rather than hypoxia per se, drives cognitive dysfunction through enhanced oxidative stress generation and suppression of antioxidant pathways. We anticipated that IH would result in greater pathway suppression, more pronounced gene expression changes, and more severe cognitive impairment compared to SH alone. To test this hypothesis, we employed both IH and SH models to investigate the differential effects of reoxygenation on cognitive function and hippocampal gene expression, aiming to elucidate the molecular mechanisms underlying OSAS-related cognitive dysfunction.

## 2. Results

### 2.1. Y-Maze Test and Passive Avoidance Test

Figure 1A shows the alternation rate in the Y-maze test. One-way analysis of variance showed no statistically significant effect of exposure conditions (F(2,46) = 1.460, *p* = 0.243). Figure 1B shows the proportion of mice that reached 300 s without entering the dark side of the chamber on day 2 of the passive avoidance test [control: 84.4% (95% CI: 68.2–93.1%), IH: 36.4% (95% CI: 15.2–64.6%), SH: 88.2% (95% CI: 65.7–96.7%)]. A chi-square test among the three groups revealed statistically significant differences between groups (χ^2^(2) = 10.719, *p* = 0.005). Pairwise comparisons using chi-square tests with Bonferroni correction showed that the IH group had a significantly lower proportion of mice reaching 300 s compared to both the control group (χ^2^(1) = 8.760, corrected *p* = 0.009) and the SH group (χ^2^(1) = 8.429, corrected *p* = 0.011). No significant difference was observed between the control and SH groups (χ^2^(1) = 0.139, *p* = 0.710).

### 2.2. Comparisons of Differentially Expressed Genes (DEGs)

To investigate the effects of IH and SH on gene expression in the mouse hippocampus, RNA sequencing (RNA-seq) was performed. Volcano plots of DEGs are shown in Figure 2A–C. Key genes of interest are labeled in the volcano plots, including *Lars2*, *Hmcn1*, *Vstm2l*, and *Rps21*, which were subsequently validated by RT-qPCR. Compared to the control group, the IH group showed 170 downregulated and 46 upregulated genes, while the SH group showed 151 downregulated and 44 upregulated genes. Direct comparison between IH and SH groups revealed 37 downregulated and 19 upregulated genes in the IH group relative to the SH group. The top 50 DEGs in each comparison, ranked by absolute log_2_ fold change (FC) magnitude, are shown in Table 1.

Figure 2D shows a Venn diagram of the DEGs. Venn diagram analysis revealed that 73 genes were shared between IH/control and SH/control comparisons, including *Pknox1* and *Klf2*. Sixteen genes were shared between IH/control and IH/SH comparisons, including *Adrb1*. Twelve genes were shared between IH/SH and SH/control comparisons, including *Dbi*. Two genes (*Cebpb* and *Gng13*) were common to all three comparisons.

### 2.3. Functional Enrichment Analysis

Gene Ontology (GO) enrichment analysis using Metascape revealed significantly altered biological processes associated with DEG patterns (Figure 3). For DEGs in the IH vs. control comparison, enriched processes included neurological functions such as learning or memory (GO: 0007611, −log_10_ (*p*) = 3.86) and brain development (GO: 0007420, −log_10_ (*p*) = 2.96), as well as response to reactive oxygen species (GO: 0000302, −log_10_ (*p*) = 3.17), blood vessel development (GO: 0001568, −log_10_ (*p*) = 3.34), and IL-17 signaling pathway (mmu04657, −log_10_ (*p*) = 2.75) (Figure 3A). In contrast, DEGs in the SH vs. control comparison showed enrichment only in response to hypoxia (GO: 0001666, −log_10_ (*p*) = 2.22) with no effects on learning or memory (Figure 3B). The IH vs. SH comparison also revealed no enrichment in learning or memory (Figure 3C).

QIAGEN Ingenuity Pathway Analysis (IPA) (QIAGEN, Hilden, Germany) identified major molecular pathways altered in each comparison. Results are displayed as positive or negative Z-scores indicating activation and inhibition, respectively. Figure 4A,B show the top 20 activated and inhibited pathways in the IH group compared to controls, respectively. Activated pathways included mitochondrial dysfunction (−log_10_ (*p*) = 19.6, Z-score = 4.838), while inhibited pathways included the KEAP1-NFE2L2 antioxidant pathway (−log_10_ (*p*) = 5.74, Z-score = −5.303). For the IH vs. SH comparison, the top 20 altered pathways are shown in Figure 4C, D, with the KEAP1-NFE2L2 pathway inhibited (−log_10_ (*p*) = 4.33, Z-score = −3.441).

### 2.4. RT-qPCR

mRNA expression changes obtained by RT-qPCR are shown in Figure 5 and Table 2. Of the 23 genes analyzed, 19 genes showed no statistically significant effects of exposure conditions, including *Adrb1*, *Foxo6*, *Trem2*, *Klf2*, *Sod3*, *Cebpb*, *Dbi*, *Manf*, *Trpc6*, *Hspa5*, *Nov*, *Basp1*, *H1fx*, *Ism1*, *Phc3*, *Pknox1*, *Rasl11a*, *Rps27*, and *Sdf2l1*.

Four genes showed significant expression changes: *Lars2* (F(2,22) = 4.376, *p* = 0.025), *Rps21* (F(2,22) = 3.870, *p* = 0.036), *Hmcn1* (F(2,22) = 9.196, *p* = 0.001), and *Vstm2l* (F(2,22) = 5.509, *p* = 0.012). Post-hoc analysis using Tukey’s HSD test revealed the following patterns: *Lars2* expression was significantly lower in the IH group compared to the SH group (*p* = 0.036) with a trend toward decrease vs. control (*p* = 0.056). *Rps21* expression was significantly lower in the SH group compared to control (*p* = 0.033). *Hmcn1* expression was significantly lower in the IH group compared to both control (*p* = 0.008) and SH groups (*p* = 0.002). *Vstm2l* expression was significantly lower in the IH group compared to control (*p* = 0.012) with a trend toward decrease vs. SH (*p* = 0.055).

## 3. Discussion

The present study demonstrates that IH induces learning or memory impairments. RNA-seq and RT-qPCR analyses indicated the involvement of the KEAP1-NFE2L2 pathway and identified novel genes potentially associated with OSAS.

Behavioral tests demonstrated that IH specifically induces cognitive dysfunction. The passive avoidance test revealed significant learning or memory impairments in the IH group, with a decreased proportion of mice exceeding the cutoff value compared to both control and SH groups, while no difference was observed between SH and control groups. These findings are consistent with previous research using similar IH conditions (minimum FiO_2_ 5%/4 min cycles/8 h daily/3 weeks) [17] and clearly indicate that intermittent, rather than sustained, hypoxia specifically impairs cognitive function. The differential sensitivity between these tests suggests that fear-motivated learning is more vulnerable to IH-induced oxidative stress than spatial working memory. This is likely because fear memory consolidation requires hippocampal neural circuit function [25], and the hippocampus is vulnerable to oxidative stress from repeated reoxygenation cycles [14]. Although the Y-maze test showed no significant differences between groups under our 4-week experimental conditions, this aligns with previous studies [17], and longer exposure durations may be required for spatial cognitive impairment [16].

RNA-seq analysis identified DEGs among groups. GO analysis of RNA-seq data revealed alterations in “learning or memory” and “response to reactive oxygen species” as key categories specifically altered in the IH group. Based on these results, RT-qPCR analysis revealed significant downregulation in four genes: *Lars2*, *Vstm2l*, *Hmcn1*, and *Rps21*. *Lars2* encodes mitochondrial leucyl-tRNA synthetase essential for mitochondrial protein synthesis and functional maintenance [26,27,28,29]. This gene has been linked to Alzheimer’s disease, with knockdown studies demonstrating direct effects on neuronal function, including shortened axon length, reduced dendritic branching, increased mitochondrial superoxide levels, and neuronal cell death, and knockout mice exhibit cognitive dysfunction, increased p-tau, and hippocampal atrophy [30]. These neuromorphological alterations suggest that *Lars2* downregulation may contribute to the cognitive deficits observed in our study. Additionally, decreased *Lars2* expression impairs mitochondrial stress responses, leading to elevated ROS levels [31]. *Vstm2l* is localized to mitochondria and involved in maintaining mitochondrial homeostasis [32]. While specific roles of *Vstm2l* in synaptic plasticity remain to be fully elucidated, mitochondrial function is essential for neuronal energy metabolism. The previous study showed that repeated cycles of hypoxia-reoxygenation increased ROS production [18]. In addition, the increase in mitochondrial ROS is associated with neuronal cell death and cognitive dysfunction [19,20]. Therefore, the downregulation of mitochondrial genes *Lars2* and *Vstm2l* suggests potential mechanisms for cognitive dysfunction that may involve impaired mitochondrial function and elevated ROS levels. *Hmcn1*, encoding an extracellular protein involved in epithelial cell junction organization [33], has been implicated in Alzheimer’s disease pathways [34]. In contrast, *Rps21*, encoding ribosomal protein S21 [35], showed downregulation specifically in the SH group. However, the preserved cognitive function in SH mice suggests that Rps21 downregulation alone may not be sufficient to cause behavioral deficits.

A key finding of this study was the significant suppression of the KEAP1-NFE2L2 pathway identified through QIAGEN IPA, with suppression observed in both IH and SH groups compared to the control group and notably stronger suppression in the IH group. The KEAP1-NFE2L2 pathway functions as a major regulator of intracellular antioxidant defense systems, playing cytoprotective roles including suppression of inflammatory signals [21], regulation of mitochondrial function [22], prevention of cell death [23], and neuroprotection [24]. Chronic IH exposure has been shown to decrease *Nrf2* expression in multiple organs, including the hippocampus [14]. In the present study, although *Nrf2* expression showed no significant differences between groups, pathway analysis revealed alterations in the KEAP1-NFE2L2 pathway. IH-induced suppression of the KEAP1-NFE2L2 pathway suggests potential attenuation of antioxidant defense and neuroprotective mechanisms, which may be associated with cognitive dysfunction. Regarding the KEAP1-NFE2L2 pathway, SH typically activates this pathway acutely but may lead to adaptive responses under chronic conditions. In contrast, IH involves repeated reoxygenation cycles that generate additional oxidative stress, resulting in greater pathway suppression than SH alone.

The coordinated downregulation of mitochondrial genes (*Lars2*, *Vstm2l*) alongside KEAP1-NFE2L2 pathway suppression suggests potential molecular pathways whereby IH exposure may affect cellular antioxidant defenses and mitochondrial function, possibly contributing to cognitive impairment. Figure 6 summarizes the proposed molecular mechanisms linking IH-induced ROS production to cognitive dysfunction through KEAP1-NFE2L2 pathway suppression and gene expression changes.

This study has several limitations. First, our behavioral and molecular analyses were limited in scope—additional cognitive tests, protein-level validation, and ROS measurements would strengthen our findings. Second, we did not examine neurobiological changes such as neuronal morphology or synaptic markers. Third, our study lacks intermediate timepoints to distinguish acute versus chronic effects of hypoxia exposure. Additionally, we did not perform correlation analysis between individual gene expression and pathway suppression scores. Future studies should include comprehensive validation approaches and multiple assessment timepoints to provide a more complete understanding of IH-induced cognitive dysfunction mechanisms.

## 4. Materials and Methods

### 4.1. Animals

Seventy-one male C57BL/6J mice (weight: 22.5 ± 2.9 g; age: 8 weeks; Japan SLC, Hamamatsu, Japan) were used for behavioral and molecular analyses. Mice were housed in groups of 5–6 per cage (30 cm length × 20 cm width × 12 cm height) under controlled environmental conditions at a temperature of 24 ± 1 °C and a 12:12 h light-dark cycle (lights on at 08:00). Mice had ad libitum access to food and water. All experimental protocols were approved by the Animal Research Committee of Showa Medical University (Tokyo, Japan) (approval code: 124040, approval date: 1 April 2024).

### 4.2. Experimental Protocol

Mice were allocated to the control group (*n* = 38), IH group (*n* = 16), and SH group (*n* = 17). The IH group mice were exposed to intermittent hypoxic loading with alternating oxygen concentrations of 10% and 21% in 2 min cycles for 8 h daily (12:00–20:00). The SH group mice were exposed to continuous hypoxic loading at 10% oxygen concentration for 8 h daily (12:00–20:00). The control group mice were housed without hypoxic exposure. Mice were maintained under these conditions for 28 days, followed by Y-maze testing on day 29, passive avoidance testing (day 1) on day 29, and passive avoidance testing (day 2) on day 30. After behavioral experiments, mice were euthanized under isoflurane anesthesia and decapitated for hippocampal collection. Hippocampi were immersed in RNA stabilization solution (RNAprotect Tissue Reagent, QIAGEN) to prevent RNA degradation, kept at 4 °C overnight, and then stored at −80 °C. Hippocampal samples were used for RNA-seq and cDNA synthesis.

### 4.3. Intermittent Hypoxia Exposure, Sustained Hypoxia Exposure

The 8 h hypoxia exposure period (12:00–20:00) during the light phase was selected based on established protocols in intermittent hypoxia research [36,37] and specifically following 8 h exposure protocols for cognitive function studies [14,17], and to mimic the inactive period in mice, corresponding to the sleep period when OSA occurs in humans. This timing aligns with the nocturnal circadian rhythm of mice, where the light phase represents their natural rest period.

The IH parameters were designed to approximate the pathophysiological patterns observed in human OSA. Clinical polysomnographic studies show that OSA patients experience desaturation-reoxygenation cycles with non-sigmoidal patterns, featuring faster reoxygenation compared to desaturation phases [38]. Our 2 min cycle protocol (70 s hypoxia, 50 s normoxia) was chosen to reflect these clinically observed temporal dynamics, providing a reasonable experimental model for studying IH-related effects. Mice in the IH group were exposed to intermittent hypoxia for 8 h daily from 12:00 to 20:00 for 28 days using a custom-made gas control delivery system…”Mice in the IH group were exposed to intermittent hypoxia for 8 h daily from 12:00 to 20:00 for 28 days using a custom-made gas control delivery system (Shibata Scientific Technology Ltd., Tokyo, Japan) with oxygen and nitrogen alternation to create alternating hypoxic and normoxic conditions, controlled by a pressure controller (Gas Cylinder Auto Changer Model 8500, WAKEN BTECH Co., Ltd., Kyoto, Japan). One cycle was defined as 120 s, consisting of a hypoxic phase (10% oxygen, 70 s) and a normoxic phase (21% oxygen, 50 s). Oxygen concentrations were continuously monitored by an oxygen analyzer (XP-3380II, Shin-Cosmos Electric Co., Ltd., Osaka, Japan) and recorded using an A/D converter (PL2604 PowerLab 4/26, ADInstruments, Dunedin, New Zealand) connected to LabChart v7 (ADInstruments).

Mice in the SH group were housed using a custom-made acrylic box (56 cm width × 45 cm height × 41 cm depth) (Kyodo International Co., Ltd., Kawasaki, Japan) with their cages placed inside. The box featured five 0.8 cm air holes: two at the upper corners of the back panel and two at the lower corners of each side panel, with a fan attached at a height of 28 cm on the left side panel that operated continuously throughout the experiment. A gas control delivery system (ProOx P110, BioSpherix, Parish, NY, USA) and a pressure controller (Gas Cylinder Auto Changer Model 8500, WAKEN BTECH Co., Ltd.) were employed to maintain hypoxic conditions using nitrogen. The system continuously monitored oxygen concentration using a built-in oxygen analyzer and featured a programmable timer function (H5S, OMRON, Kyoto, Japan). Mice were exposed to sustained hypoxia at 10% oxygen concentration for 8 h daily (12:00–20:00) over 28 days. During the remaining period (20:00–12:00), atmospheric oxygen concentration (21%) was maintained using an air pump (AP-30P, Yasunaga Air Pump Co., Ltd., Tokyo, Japan).

Both gas delivery systems were validated before experiments and underwent regular calibration to ensure accurate gas circulation and maintain stable concentrations.

### 4.4. Y-Maze Test

Based on previous studies [16,17,39], mouse short-term spatial working memory was evaluated using the Y-maze test. Forty-nine mice (control group: *n* = 17, IH group: *n* = 16, SH group: *n* = 17) were analyzed, with one mouse from the IH group excluded due to deviation from the Y-maze test during the experiment. The maze consisted of three identical arms radiating from a central area at 120° angles to each other, with dimensions of 40 cm length × 12 cm height, 3 cm width at the floor expanding to 10 cm width at the ceiling. Arm entry was defined as all four paws completely entering an arm. Mice explored the maze for 8 min while the number of arm entries was recorded by an overhead video camera (MX Brio C1100PG, Logitech International S.A., Lausanne, Switzerland) and analyzed using behavioral analysis software (SMART V 3.0, Bio Research Center Co., Ltd., Nagoya, Japan). The percentage of spontaneous alternation was calculated as: %Alternations = (number of triads/(N − 2)) × 100, where N = total number of entries.

### 4.5. Passive Avoidance Test

Sample size calculations were based on standard guidelines for χ^2^ analysis. For detecting a large effect size (Cohen’s h ≥ 0.7) with α = 0.05 and power = 0.80, the minimum required sample size is approximately *n* = 16 per group. Based on previous studies [17,40], mouse memory and learning ability were evaluated using a two-compartment step-through passive avoidance apparatus (MPB-M030, Melquest Co., Toyama, Japan). Seventy-one mice (control group: *n* = 38, IH group: *n* = 16, SH group: *n* = 17) were initially used. The apparatus consisted of a bright compartment (10.0 × 18.0 × 14.5 cm) and a dark compartment (18.0 × 18.0 × 14.5 cm) separated by a wall with a guillotine door. The bright compartment was illuminated at 145 lx.

During the training phase, mice were placed in the bright compartment for 20 s before the guillotine door was opened to allow entry into the dark compartment. When mice entered the dark compartment, the guillotine door was closed, and an electric shock (0.3 mA) was delivered for 3 s. The 0.3 mA shock intensity was selected based on manufacturer recommendations (Melquest Co.) and is consistent with established protocols using 0.3 mA for 3 s in passive avoidance testing [40], ensuring reliable learning responses while maintaining appropriate animal welfare standards. Testing was performed 24 h after training. Mice were placed in the bright compartment for 20 s before the guillotine door was opened. The latency to enter the dark compartment was recorded for up to 300 s. Mice that did not enter the dark compartment after 300 s were considered to have retained memory, and this proportion was used for analysis. Due to experimental protocol errors, 11 mice (6 from the control group and 5 from the IH group) that received multiple electrical stimulations were excluded from the analysis, resulting in final group sizes of *n* = 32 (control), *n* = 11 (IH), and *n* = 17 (SH).

### 4.6. RNA Sequencing

Total RNA was extracted from hippocampi using the RNeasy^®^ Plus Universal Mini Kit (QIAGEN) following the manufacturer’s protocol. Total RNA concentration and purity were determined based on the ratio of absorbance at 260 nm and 280 nm using a NanoDrop One Spectrophotometer (Thermo Fisher Scientific, Waltham, MA, USA). RNA samples with A260/280 ratios ≥ 1.8 and A260/230 ratios ≥ 1.8 were considered acceptable for further analysis. For RNA-seq analysis, 5 samples each from the IH and SH groups and 4 samples from the control group were submitted to Rhelixa Co., Ltd. (Tokyo, Japan). Control samples were prepared from mice exposed to atmospheric conditions under identical experimental conditions (same mouse strain, age, and housing duration) in a preliminary experiment. RNA integrity was assessed by Rhelixa Co., Ltd., and only samples with RNA Integrity Number ≥ 8.0 were used for sequencing. Library preparation was performed using the NEBNext Ultra II Directional RNA Library Prep Kit with NEBNext Poly(A) mRNA Magnetic Isolation Module for poly-A selection (New England Biolabs, Ipswich, MA, USA). Sequencing was conducted on an Illumina NovaSeq 6000 platform (Illumina, San Diego, CA, USA), generating paired-end 150 bp reads with an average sequencing depth of 26.7 million reads per sample.

In the primary analysis, adapter sequences and low-quality bases in paired-end reads were removed with fastp (version 0.23.4). Filtered paired-end reads were mapped to the mouse reference genome (GRCm39) by HISAT2 (version 2.2.1) and expression levels were quantified by StringTie (version 2.2.1). The read count values of the gene expression data (raw signal) were subjected to appropriate processing by Subio Platform (Subio Inc, Nagoya, Japan), including binary log transformation, global normalization by 80th percentile, low signal cutoff (read counts < 40), missing value completion (log_2_ 32), correction by control group values (Processed signal), averaging and ratio calculation to the control group by exponential function (FC). Specifically, among the read gene expression data (40,015 genes), only genes for which the raw signal in each group was above the low signal cutoff value for protein-coding genes were extracted (13,733 genes) and further filtered to genes with a processed signal ≥ |0.25| (7756 genes).

For secondary analysis, functional analysis was performed using Metascape to identify enriched biological pathways and processes [41]. DEG lists were analyzed with thresholds of FC > 1.5 or < 0.67 and significance levels with *p* < 0.05. Furthermore, pathway analysis was conducted using QIAGEN IPA to examine molecular networks. Of the 7756 genes, 38 genes that were not mapped within the QIAGEN IPA were excluded, and of the remaining 7718 genes, those meeting *p* < 0.05 were analyzed. Data were considered significant with thresholds of −log_10_ (*p*) > 1.3 and |Z-score| > 2.0. The sequence data were deposited in the Gene Expression Omnibus database (https://www.ncbi.nlm.nih.gov/geo, accessed on 25 July 2025; accession no. GSE 299437).

### 4.7. RT-qPCR

For RT-qPCR analysis, total RNA samples from the control group included 9 samples extracted in this experiment, while the IH and SH groups used 8 samples from each group. cDNA was synthesized from total RNA using PrimeScript™ RT Master Mix (Takara Bio Inc., Kusatsu, Japan) according to the manufacturer’s protocol.

Target genes were selected based on several criteria. Based on GO analysis results, genes with read counts >100 that have been reported to be associated with cognitive function or neural processes within the “learning or memory” and “response to reactive oxygen species” categories were selected (*Adrb1* [42], *Foxo6* [43], *Trem2* [44], *Klf2* [45], *Sod3* [46]). Additionally, three categories of genes were included: genes previously reported to be associated with cognitive function or neural processes but not identified in the GO analysis (*Cebpb* [47], *Dbi* [48], *Lars2* [30], *Manf* [49], *Trpc6* [50]), genes with high read counts (>1500) but no existing literature reports (*Hspa5*, *Nov*, *Rps21*), and genes with large FC (>2.0 or <0.5) (*Basp1*, *H1fx*, *Hmcn1*, *Ism1*, *Phc3*, *Pknox1*, *Rasl11a*, *Rps27*, *Sdf2l1*, *Vstm2l*).

Probe details are provided in Table 3. All probes contained 6-FAM at the 5′ end, ZEN quencher internally, and Iowa Black FQ quencher at the 3′ end and were synthesized through a PrimeTime Mini qPCR Assay (Integrated DNA Technologies, Inc., Coralville, IA, USA). Quantitative PCR was performed in a total volume of 10 μL with PrimeTime^®^ Gene Expression Master Mix (Integrated DNA Technologies, Inc., Coralville, IA, USA) and QuantStudio™ 5 Real-Time PCR system (Thermo Fisher Scientific, Waltham, MA, USA) according to the manufacturer’s instructions. The PCR conditions were as follows: initial denaturation at 95 °C for 3 min, followed by 45 cycles of 95 °C for 15 s and 60 °C for 30 s.

The relative expression levels of each gene were determined by the ΔΔCt method using *Gapdh* as an internal control. *Gapdh* was selected as the reference gene based on its established stability in intermittent hypoxia studies. Previous validation studies using RT-qPCR have demonstrated *Gapdh* stability in chronic intermittent hypoxia models of sleep apnea in both brain tissue [51] and cardiac tissue [52], supporting its use as an appropriate internal control for this experimental paradigm.

### 4.8. Statistical Analysis

Statistical analyses were performed using JMP Pro ver.17.0.0 (JMP Statistical Discovery LLC., Cary, NC, USA). All data are presented as mean ± standard error. For Y-maze test alternation behavior rates and hippocampal mRNA levels, one-way analysis of variance (ANOVA) followed by Tukey’s HSD test was used for multiple group comparisons. For the passive avoidance test, the proportion of mice exceeding the cutoff value (300 s) was compared using the chi-square test. Statistical significance was set at *p* < 0.05.

## 5. Conclusions

In this study, we demonstrated that IH induces learning or memory impairments using a mouse model of OSAS. RNA-seq and RT-qPCR analyses revealed three genes (*Lars2*, *Hmcn1*, and *Vstm2l*) that showed specific downregulation in the IH group and confirmed the involvement of the KEAP1-NFE2L2 antioxidant pathway. These findings provide insights into the molecular basis of cognitive dysfunction in OSAS.

## Figures and Tables

**Figure 1 ijms-26-07495-f001:**
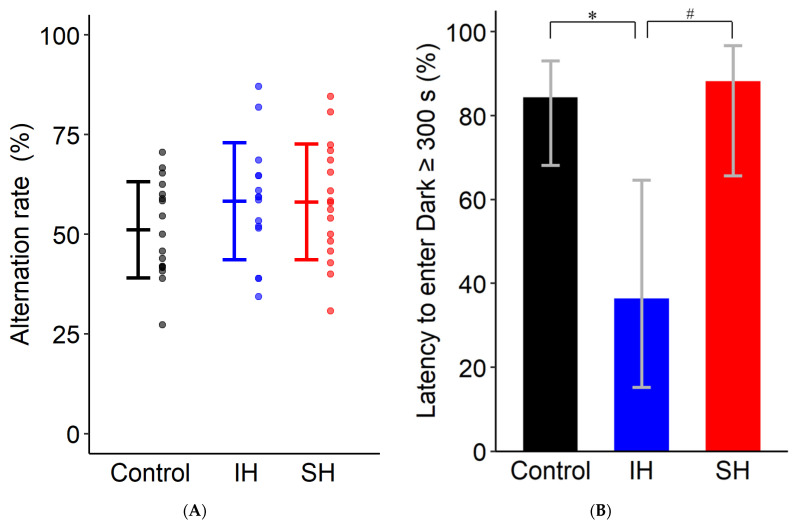
Behavioral test results. (**A**) Alternation rate in the Y-maze test. No statistically significant differences were observed among the control, IH, and SH groups (F(2,46) = 1.460, *p* = 0.243). (**B**) Percentage of mice with dark chamber entry latency ≥ 300 s in the passive avoidance test. The IH group showed a significantly lower percentage compared to the control group (* *p* = 0.009, Bonferroni-corrected). The IH group also showed a significantly lower percentage compared to the SH group (# *p* = 0.011, Bonferroni-corrected). No significant difference was observed between the control and SH groups (χ^2^(1) = 0.139, *p* = 0.710). Data are presented as mean ± standard error (**A**); control: *n* = 17, IH: *n* = 15, SH: *n* = 17) or as percentage with 95% binomial confidence intervals (**B**; control: *n* = 32, IH: *n* = 11, SH: *n* = 17). Panel (**B**) presents binary categorical data (success/failure for individual mice), which differs from the continuous data presentation in other panels. Statistical significance for (**A**) was determined by one-way ANOVA, and for (**B**) by chi-square test with Bonferroni correction. Statistical significance was set at *p* < 0.05. * *p* < 0.05, IH vs. control; # *p* < 0.05, IH vs. SH. IH, intermittent hypoxia; SH, sustained hypoxia.

**Figure 2 ijms-26-07495-f002:**
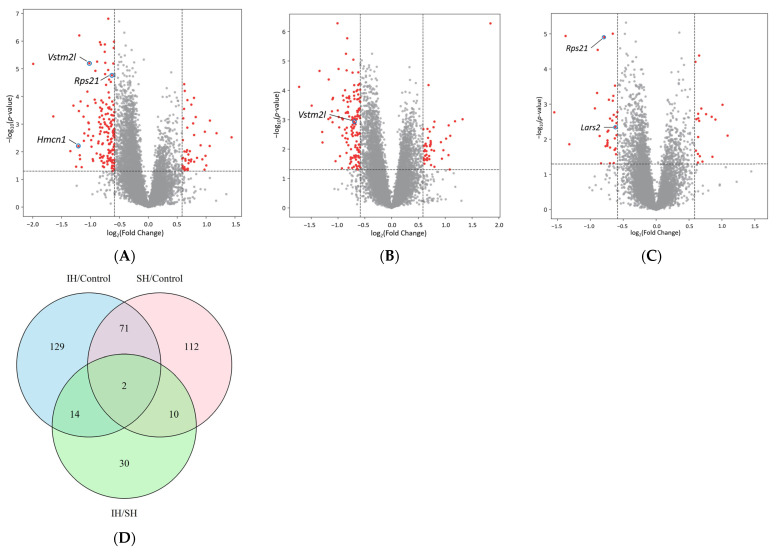
Differentially expressed genes (DEGs) analysis (control: *n* = 4, IH and SH: *n* = 5 each). (**A**–**C**) Volcano plots showing gene expression changes between different oxygen exposure conditions. (**A**) IH vs. control, (**B**) SH vs. control, (**C**) IH vs. SH. The *X*-axis shows log_2_ fold change (FC); the *Y*-axis shows −log_10_ (*p*-value). Red dots represent DEGs. Statistical significance was set at |log_2_ (FC)| > 0.58 and −log_10_ (*p*) > 1.3. Key genes of interest (*Lars2*, *Hmcn1*, *Vstm2l*, *Rps21*) are labeled on the volcano plots. (**D**) Venn diagram of DEGs among the three comparisons. Seventy-three genes were shared between IH vs. control and SH vs. control comparisons, sixteen genes between IH vs. control and IH vs. SH comparisons, and twelve genes between IH vs. SH and SH vs. control comparisons. Two genes were common to all three comparisons. IH, intermittent hypoxia; SH, sustained hypoxia.

**Figure 3 ijms-26-07495-f003:**
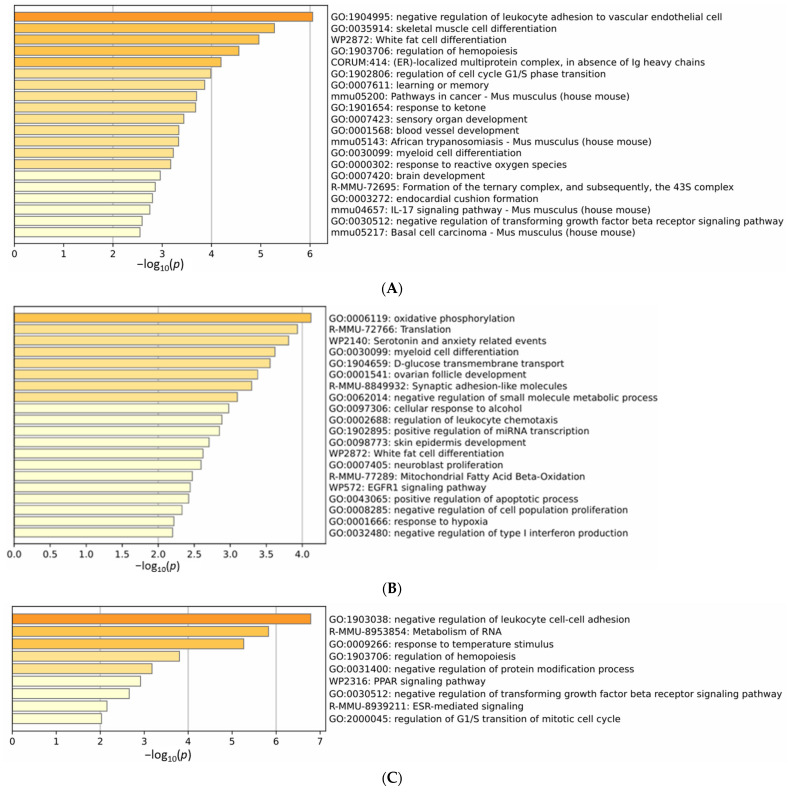
Gene Ontology (GO) analysis of biological processes (control: *n* = 4, IH and SH: *n* = 5 each). (**A**) GO analysis results for DEGs in the IH vs. control comparison. The *X*-axis shows −log10 (*p*) values. In addition to neurological processes such as learning or memory and brain development, response to reactive oxygen species and blood vessel development showed significant enrichment. (**B**) GO analysis results for DEGs in the SH vs. control comparison. Enrichment in response to hypoxia was observed, but no significant enrichment in learning, memory, or neurological processes was found. (**C**) GO analysis results for DEGs in the IH vs. SH comparison. No significant enrichment in neurological processes was observed. Statistical significance was set at *p* < 0.05.

**Figure 4 ijms-26-07495-f004:**
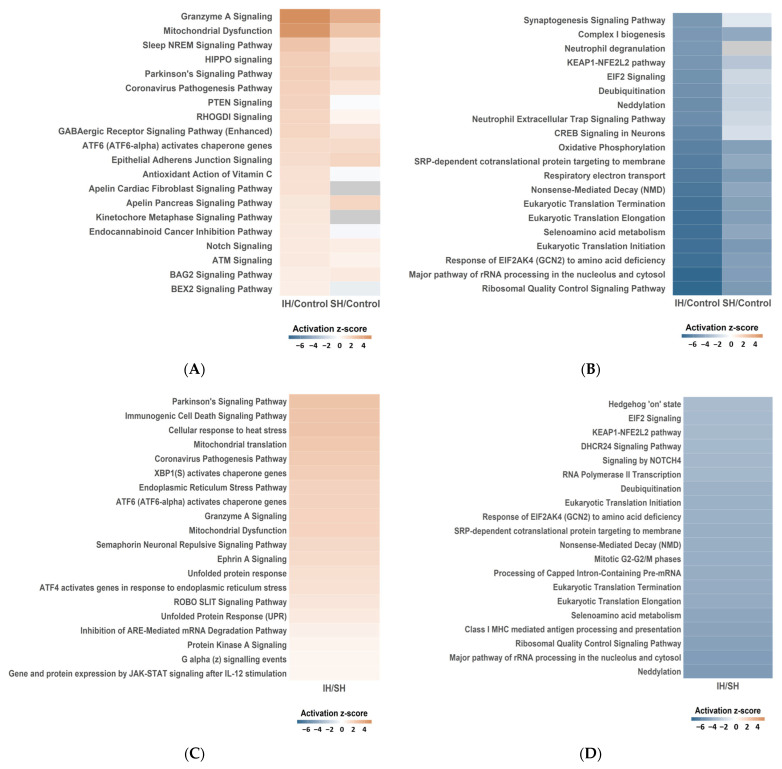
QIAGEN Ingenuity Pathway Analysis (QIAGEN IPA) results showing altered molecular pathways (control: *n* = 4, IH and SH: *n* = 5 each). (**A**) Major molecular pathways activated in the IH group compared to the control group. Pathways such as mitochondrial dysfunction were activated. (**B**) Major molecular pathways inhibited in the IH group compared to the control group. The KEAP1-NFE2L2 antioxidant pathway was inhibited. (**C**) Major molecular pathways activated in the IH group compared to the SH group. (**D**) Major molecular pathways inhibited in the IH group compared to the SH group. The KEAP1-NFE2L2 antioxidant pathway was inhibited. The *X*-axis shows −log_10_ (*p*-value) and the color scale shows activation Z-score, with red indicating activation and blue indicating inhibition. Statistical significance was set at −log_10_ (*p*) > 1.3 and |Z-score| > 2.0. IH, intermittent hypoxia; SH, sustained hypoxia.

**Figure 5 ijms-26-07495-f005:**
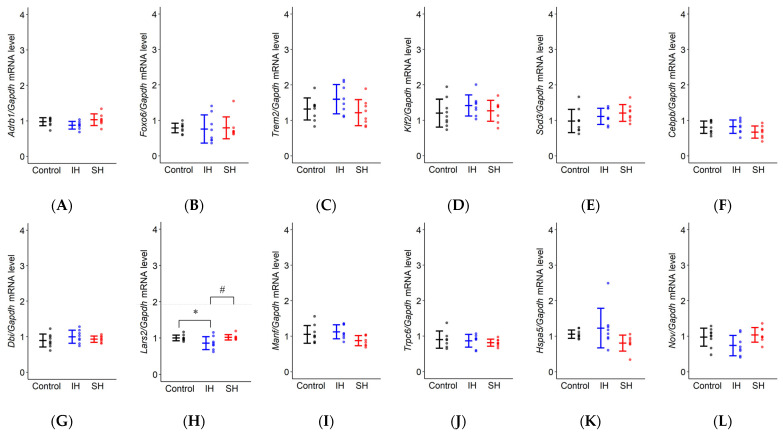
*Adrb1, Foxo6, Trem2, Klf2, Sod3, Cebpb, Dbi, Lars2, Manf, Trpc6, Hspa5, Nov, Rps21, Basp1, H1fx, Hmcn1, Ism1, Phc3, Pknox1, Rasl11a, Rps27, Sdf2l1,* and *Vstm2l.* (**A**–**W**) mRNA expression in mouse hippocampus. Values are presented as mean ± standard error (control: *n* = 9, IH and SH: *n* = 8 each). (**H**) *Lars2* mRNA in the IH group showed a significant decrease compared to the SH group (# *p* = 0.036) and a trend towards decrease compared to the control group (*p* = 0.056). (**M**) *Rps21* mRNA in the SH group showed a significant decrease compared to the control group († *p* = 0.033). (**P**) *Hmcn1* mRNA in the IH group showed a significant decrease compared to both the control group (* *p* = 0.008) and the SH group (# *p* = 0.002). (**W**) *Vstm2l* mRNA in the IH group showed a significant decrease compared to the control group (* *p* = 0.012) and a trend toward decrease compared to the SH group (*p* = 0.055). Statistical analysis was performed using one-way ANOVA followed by Tukey’s HSD test. * *p* < 0.05 vs. control; # *p* < 0.05, IH vs. SH; † *p* < 0.05, SH vs. control. Statistical significance was set at *p* < 0.05. IH, intermittent hypoxia; SH, sustained hypoxia.

**Figure 6 ijms-26-07495-f006:**
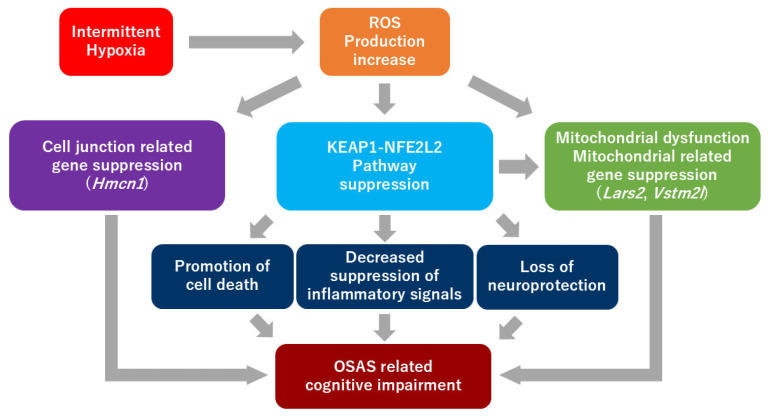
Proposed molecular mechanisms of cognitive dysfunction induced by intermittent hypoxia. IH leads to increased ROS production, which suppresses the KEAP1-NFE2L2 antioxidant pathway. This central pathway suppression results in multiple downstream effects including decreased suppression of inflammatory signals [21], promotion of cell death [23], and loss of neuroprotection [24]. Additionally, IH-induced ROS production causes suppression of mitochondrial-related genes (*Lars2*, *Vstm2l*) and cell junction-related genes (*Hmcn1*), leading to mitochondrial dysfunction [22]. These molecular pathways converge to ultimately cause OSAS-related cognitive impairment. IH, intermittent hypoxia; ROS, reactive oxygen species.

**Table 1 ijms-26-07495-t001:** The top 50 DEGs in each comparison, ranked by absolute log_2_ FC magnitude.

IH/Control			SH/Control			IH/SH		
Gene	log_2_FC	*p*-Value	Gene	log_2_FC	*p*-Value	Gene	log_2_FC	*p*-Value
*Cebpb*	−1.993	6.666 × 10^−6^	*Phc3*	1.843	5.262 × 10^−7^	*Phc3*	−1.545	1.708 × 10^−3^
*Fos*	−1.644	5.295 × 10^−4^	*Gng13*	−1.726	7.609 × 10^−5^	*Cebpb*	−1.373	1.165 × 10^−5^
*Plin4*	1.441	3.014 × 10^−3^	*Npas4*	−1.492	3.315 × 10^−4^	*H3c14*	−1.317	1.381 × 10^−2^
*Npas4*	−1.349	9.500 × 10^−3^	*Cysrt1*	−1.346	2.177 × 10^−5^	*Map3k6*	1.086	7.855 × 10^−3^
*C2cd4b*	−1.296	2.150 × 10^−4^	*Plin4*	1.323	9.619 × 10^−4^	*Hspb1*	1.009	1.044 × 10^−3^
*Rps27rt*	−1.249	3.471 × 10^−2^	*Egr4*	−1.295	2.606 × 10^−3^	*Dynlt1f*	−0.929	1.321 × 10^−3^
*Hmcn1*	−1.209	6.247 × 10^−3^	*Pcdha11*	−1.290	5.937 × 10^−3^	*Gng13*	0.901	2.744 × 10^−3^
*Cysrt1*	−1.201	3.383 × 10^−4^	*Ankub1*	1.182	1.129 × 10^−3^	*Rpl17-ps8*	−0.895	4.822 × 10^−4^
*H1f10*	−1.195	6.248 × 10^−7^	*C2cd4b*	−1.181	9.489 × 10^−4^	*Rpl39*	−0.886	2.877 × 10^−5^
*Hbb-bt*	−1.187	1.340 × 10^−2^	*Enho*	−1.174	4.272 × 10^−5^	*Snurf*	−0.857	8.045 × 10^−3^
*Arc*	−1.181	1.848 × 10^−2^	*Ier2*	−1.145	8.485 × 10^−4^	*Moap1*	0.856	3.136 × 10^−2^
*Pknox1*	1.181	2.139 × 10^−3^	*Junb*	−1.114	5.181 × 10^−4^	*Phospho1*	0.844	2.287 × 10^−3^
*Junb*	−1.172	1.515 × 10^−4^	*Fos*	−1.104	2.005 × 10^−4^	*Gnpnat1*	−0.836	4.827 × 10^−2^
*Egr2*	−1.153	3.605 × 10^−2^	*Gm1673*	−1.103	1.217 × 10^−3^	*Rps21*	−0.788	1.248 × 10^−5^
*Btg2*	−1.121	5.394 × 10^−4^	*Slc18a2*	1.093	3.518 × 10^−3^	*Tnfsf10*	−0.766	1.568 × 10^−2^
*Pla2g4e*	−1.115	4.045 × 10^−3^	*Btg2*	−1.093	1.799 × 10^−4^	*Hspa1b*	0.759	1.929 × 10^−3^
*Egr4*	−1.107	2.171 × 10^−3^	*Zic4*	1.084	4.950 × 10^−2^	*Gm13304*	−0.740	1.380 × 10^−2^
*Sdf2l1*	1.067	7.463 × 10^−4^	*Inhba*	1.064	1.554 × 10^−3^	*Gm13306*	−0.740	1.525 × 10^−2^
*Slc18a2*	1.062	6.197 × 10^−3^	*Gm44126*	1.060	1.597 × 10^−2^	*Rpl17*	−0.738	1.073 × 10^−2^
*Adrb1*	−1.055	6.714 × 10^−5^	*Arc*	−1.047	2.575 × 10^−2^	*Gm10591*	−0.737	6.057 × 10^−3^
*Tmem238*	−1.049	2.205 × 10^−4^	*Pknox1*	1.016	4.399 × 10^−3^	*Ccl21b*	−0.736	1.243 × 10^−2^
*Ier2*	−1.042	2.628 × 10^−3^	*Mrpl52*	−1.010	5.217 × 10^−7^	*Gm20498*	−0.724	5.646 × 10^−3^
*Dusp1*	−1.038	8.774 × 10^−4^	*Rasl11a*	−1.005	4.835 × 10^−4^	*Hmgcs2*	−0.716	7.510 × 10^−4^
*Rasl11a*	−1.031	9.329 × 10^−4^	*Pcsk1n*	−0.990	1.870 × 10^−5^	*Rnasek*	0.704	4.290 × 10^−2^
*Ism1*	−1.025	1.550 × 10^−3^	*Tmem238*	−0.987	8.084 × 10^−4^	*Dcst1*	−0.701	4.809 × 10^−2^
*Vstm2l*	−1.021	6.329 × 10^−6^	*Wnt9a*	−0.979	1.821 × 10^−3^	*Pcdha3*	−0.699	2.567 × 10^−3^
*Ankub1*	1.004	3.126 × 10^−2^	*Tnfaip6*	0.965	1.090 × 10^−2^	*U2af1l4*	−0.698	1.711 × 10^−2^
*Zfp708*	0.990	1.405 × 10^−2^	*Oxld1*	−0.963	1.739 × 10^−2^	*Hexim2*	0.687	1.335 × 10^−3^
*Tusc1*	−0.989	3.795 × 10^−3^	*Dnah12*	0.938	2.121 × 10^−2^	*Gstp-ps*	−0.665	2.933 × 10^−3^
*Adamts16*	−0.983	6.563 × 10^−3^	*Apold1*	−0.928	4.507 × 10^−2^	*2300009A05Rik*	0.664	3.020 × 10^−2^
*Inhba*	0.980	1.889 × 10^−3^	*Mrpl54*	−0.926	8.988 × 10^−4^	*Itm2a*	−0.657	9.970 × 10^−6^
*Lamc2*	0.973	4.439 × 10^−2^	*Prrg1*	0.919	5.807 × 10^−3^	*Adrb1*	−0.657	8.013 × 10^−4^
*Pcdhgb5*	−0.961	1.859 × 10^−2^	*Phospho1*	−0.918	3.473 × 10^−3^	*Lfng*	0.654	2.993 × 10^−3^
*Iqschfp*	−0.955	2.146 × 10^−2^	*Gm2423*	−0.911	9.762 × 10^−4^	*Dbi*	0.654	4.166 × 10^−5^
*Ccl21b*	−0.955	2.696 × 10^−3^	*S100a13*	−0.910	1.517 × 10^−4^	*Rps28*	−0.648	5.831 × 10^−4^
*Peg10*	0.935	1.883 × 10^−2^	*Rps29*	−0.896	2.184 × 10^−4^	*Scrg1*	−0.642	2.020 × 10^−3^
*Hba-a2*	−0.928	1.310 × 10^−4^	*Ly6h*	−0.872	2.158 × 10^−3^	*Adamts16*	−0.642	4.729 × 10^−2^
*Gm13889*	−0.916	1.187 × 10^−5^	*Trnp1*	−0.868	9.911 × 10^−5^	*Hspa1a*	0.641	2.541 × 10^−3^
*Sap30l*	−0.913	1.711 × 10^−4^	*Klf2*	−0.864	4.240 × 10^−3^	*Mrpl54*	0.639	1.883 × 10^−3^
*Plekhg4*	0.905	5.845 × 10^−3^	*Myl6b*	−0.856	5.866 × 10^−4^	*Ptpn6*	0.639	8.596 × 10^−3^
*Gm10591*	−0.894	1.155 × 10^−2^	*Sap30l*	−0.852	5.689 × 10^−6^	*Peg10*	0.636	4.555 × 10^−2^
*Fbxl9*	−0.886	5.532 × 10^−6^	*Erf*	−0.844	5.686 × 10^−4^	*Rtl3*	0.635	2.598 × 10^−2^
*Wnt9a*	−0.866	4.959 × 10^−3^	*Tmem160*	−0.843	2.832 × 10^−3^	*Fancd2*	−0.631	1.993 × 10^−2^
*Frat2*	−0.861	4.246 × 10^−3^	*Sox18*	−0.842	1.348 × 10^−2^	*Ror1*	−0.626	9.413 × 10^−3^
*Sts*	−0.855	8.503 × 10^−4^	*Rpl38*	−0.840	2.093 × 10^−5^	*Otogl*	−0.624	1.961 × 10^−2^
*Insm1*	−0.844	5.657 × 10^−4^	*Tmem256*	−0.840	1.783 × 10^−4^	*Rpl22l1*	−0.624	2.989 × 10^−4^
*Hspb1*	0.843	1.663 × 10^−2^	*Tead3*	−0.840	5.864 × 10^−3^	*Lars2*	−0.619	4.500 × 10^−3^
*Tomm6*	−0.838	2.268 × 10^−3^	*Zfp524*	−0.836	1.869 × 10^−3^	*Nadsyn1*	−0.614	1.815 × 10^−2^
*Arf6*	−0.837	1.101 × 10^−6^	*Hcfc1r1*	−0.829	1.688 × 10^−6^	*Cdk2ap2*	0.611	2.685 × 10^−3^
*Pgam2*	−0.836	1.671 × 10^−2^	*Chchd10*	−0.822	2.469 × 10^−4^	*Ppara*	−0.608	2.653 × 10^−2^

**Table 2 ijms-26-07495-t002:** RT-qPCR results for mRNA expression levels.

Gene	F Value	LS Mean	IH vs. Control	SH vs. Control	IH vs. SH
*Adrb1*	F(2,22) = 2.915, *p* = 0.075	Control: 0.977 ± 0.064 IH: 0.877 ± 0.066 SH: 1.033 ± 0.066	*p* = 0.277	*p* = 0.662	*p* = 0.066
*Foxo6*	F(2,22) = 0.029, *p* = 0.971	Control: 0.786 ± 0.144 IH: 0.758 ± 0.148 SH: 0.791 ± 0.148	*p* = 0.979	*p* = 0.999	*p* = 0.973
*Trem2*	F(2,22) = 2.343, *p* = 0.120	Control: 1.321 ± 0.176 IH: 1.597 ± 0.181 SH: 1.218 ± 0.181	*p* = 0.282	*p* = 0.831	*p* = 0.116
*Klf2*	F(2,22) = 0.899, *p* = 0.421	Control: 1.202 ± 0.163 IH: 1.417 ± 0.168 SH: 1.269 ± 0.168	*p* = 0.400	*p* = 0.912	*p* = 0.656
*Sod3*	F(2,22) = 1.495, *p* = 0.246	Control: 0.984 ± 0.132 IH: 1.114 ± 0.135 SH: 1.209 ± 0.135	*p* = 0.588	*p* = 0.222	*p* = 0.765
*Cebpb*	F(2,22) = 1.808, *p* = 0.188	Control: 0.808 ± 0.087 IH: 0.823 ± 0.089 SH: 0.670 ± 0.089	*p* = 0.984	*p* = 0.271	*p* = 0.223
*Dbi*	F(2,22) = 0.930, *p* = 0.409	Control: 0.892 ± 0.078 IH: 0.997 ± 0.080 SH: 0.928 ± 0.080	*p* = 0.384	*p* = 0.891	*p* = 0.668
*Lars2*	F(2,22) = 4.376, * *p* = 0.025	Control: 0.999 ± 0.058 IH: 0.857 ± 0.059 SH: 1.016 ± 0.059	*p* = 0.056	*p* = 0.954	* *p* = 0.036
*Manf*	F(2,22) = 3.191, *p* = 0.061	Control: 1.051 ± 0.099 IH: 1.123 ± 0.102 SH: 0.872 ± 0.102	*p* = 0.753	*p* = 0.194	*p* = 0.057
*Trpc6*	F(2,20) = 0.452, *p* = 0.643	Control: 0.900 ± 0.094 IH: 0.866 ± 0.090 SH: 0.812 ± 0.090	*p* = 0.932	*p* = 0.624	*p* = 0.822
*Hspa5*	F(2,21) = 2.733, *p* = 0.088	Control: 1.057 ± 0.177 IH: 1.226 ± 0.182 SH: 0.802 ± 0.182	*p* = 0.591	*p* = 0.340	*p* = 0.073
*Nov*	F(2,22) = 2.310, *p* = 0.123	Control: 0.973 ± 0.120 IH: 0.738 ± 0.123 SH: 0.954 ± 0.123	*p* = 0.145	*p* = 0.987	*p* = 0.207
*Rps21*	F(2,22) = 3.870, * *p* = 0.036	Control: 1.029 ± 0.111 IH: 0.826 ± 0.114 SH: 0.729 ± 0.114	*p* = 0.182	* *p* = 0.033	*p* = 0.676
*Basp1*	F(2,22) = 0.151, *p* = 0.861	Control: 0.792 ± 0.173 IH: 0.741 ± 0.178 SH: 0.697 ± 0.178	*p* = 0.953	*p* = 0.849	*p* = 0.967
*H1fx*	F(2,22) = 0.110, *p* = 0.897	Control: 0.912 ± 0.122 IH: 0.913 ± 0.125 SH: 0.963 ± 0.125	*p* = 1.000	*p* = 0.909	*p* = 0.916
*Hmcn1*	F(2,22) = 9.196, * *p* = 0.001	Control: 1.271 ± 0.126 IH: 0.849 ± 0.130 SH: 1.373 ± 0.130	* *p* = 0.008	*p* = 0.703	* *p* = 0.002
*Ism1*	F(2,22) = 0.204, *p* = 0.817	Control: 1.122 ± 0.130 IH: 1.044 ± 0.134 SH: 1.110 ± 0.134	*p* = 0.821	*p* = 0.995	*p* = 0.875
*Phc3*	F(2,22) = 0.423, *p* = 0.660	Control: 1.286 ± 0.177 IH: 1.179 ± 0.182 SH: 1.344 ± 0.182	*p* = 0.819	*p* = 0.943	*p* = 0.643
*Pknox1*	F(2,22) = 1.774, *p* = 0.193	Control: 1.030 ± 0.109 IH: 0.840 ± 0.112 SH: 1.010 ± 0.112	*p* = 0.212	*p* = 0.980	*p* = 0.306
*Rasl11a*	F(2,22) = 1.129, *p* = 0.342	Control: 0.988 ± 0.077 IH: 1.057 ± 0.079 SH: 0.938 ± 0.079	*p* = 0.652	*p* = 0.795	*p* = 0.312
*Rps27*	F(2,22) = 0.640, *p* = 0.537	Control: 1.067 ± 0.084 IH: 0.993 ± 0.086 SH: 0.984 ± 0.086	*p* = 0.570	*p* = 0.642	*p* = 0.993
*Sdf2l1*	F(2,22) = 2.764, *p* = 0.085	Control: 1.464 ± 0.286 IH: 1.327 ± 0.295 SH: 0.818 ± 0.295	*p* = 0.882	*p* = 0.084	*p* = 0.218
*Vstm2l*	F(2,22) = 5.509, * *p* = 0.012	Control: 0.840 ± 0.100 IH: 0.521 ± 0.103 SH: 0.775 ± 0.103	* *p* = 0.012	*p* = 0.799	*p* = 0.055

Asterisks indicate statistical significance: * *p* < 0.05. Statistical analysis was performed using one-way ANOVA. Tukey’s HSD test was conducted to examine differences between groups.

**Table 3 ijms-26-07495-t003:** Probes used for RT-qPCR analysis.

Gene	Forward Primer	Reverse Primer	Probe Sequence
*Adrb1*	5′-GTTTACTCAAGACCGAAAGCAG-3′	5′-CACTCTCCCAACTCCTCCTAA-3′	5′-/56-FAM/ATGCAAAGC/ZEN/CCACAGATCTATCGAATCA/3IABkFQ/-3′
*Basp1*	5′-CCTTTGCTGAGCGACCA-3′	5′-AGCTTGCCTCCCATCTTG-3′	5′-/56-FAM/TGAGCGCGG/ZEN/TGCCTCCAA/3IABkFQ/-3′
*Cebpb*	5′-GTTTCGGGACTTGATGCAATC-3′	5′-CCGCAGGAACATCTTTAAGTGA-3′	5′-/56-FAM/ACACGGGAC/ZEN/TGACGCAACACA/3IABkFQ/-3′
*Dbi*	5′-CATCTACAGTCACTTCAAACAAGC-3′	5′-ACATAGGTCTTCATGGCACTT-3′	5′-/56-FAM/ACTCGTGGA/ZEN/ACAAGCTGAAAGGAC/3IABkFQ/-3′
*Foxo6*	5′-AGGATAAAGCGACAGCAAC-3′	5′-CACCATGAACTCTTGCCAGT-3′	5’-/56-FAM/AGAACTCCA/ZEN/TTCGGCACAACCTGT/3IABkFQ/-3’
*H1fx*	5’-CAGGAAGGTGGCATGGTT-3’	5’-TGCAGTAGCGTATCGTTCTG-3’	5’-/56-FAM/AGCGCCCGG/ZEN/ATAGAGTACTTGAGG/3IABkFQ/-3’
*Hmcn1*	5’-AGTAAGCACTACAGCCTTCAAG-3’	5’-GCACGTCATAGAGGTAGAACTG-3’	5’-/56-FAM/CCACCGAAT/ZEN/ATGGACAACGCAATGG/3IABkFQ/-3’
*Hspa5*	5’-AGAGTTCTTCAATGGCAAGGAG-3’	5′-ATCAAGCAGTACCAGATCACC-3′	5′-/56-FAM/ACAGCGGCA/ZEN/CCATAGGCTACAG/3IABkFQ/-3′
*Ism1*	5′-GATGACAGCAACTTCCTCAGT-3′	5′-AGACAGACCAGAGACTCCAAT-3′	5′-/56-FAM/AGAGCAGCC/ZEN/AGAGTATGATTCCACAGA/3IABkFQ/-3′
*Klf2*	5′-GCAAGACCTACACCAAGAGC-3′	5′-CTTCCAGCCGCATCCTTC-3′	5′-/56-FAM/TGCGTACAC/ZEN/ACACAGGTGAGAAGC/3IABkFQ/-3′
*Lars2*	5′-GTTCTATGCACGATTCCTCAGT-3′	5′-ATGGAAGGCGGAATGTCTG-3′	5′-/56-FAM/AAGGTTCCC/ZEN/TGTGCTTCACCATCTT/3IABkFQ/-3′
*Manf*	5′-TCACATTTTCACCAGCCACTA-3′	5′-CTTCGACACCTCATTGATGATCT-3′	5′-/56-FAM/ACCGATTCT/ZEN/CTTTGCCTCTTGCTTCA/3IABkFQ/-3′
*Nov*	5′-AGATGAGACCCTGTGACCAG-3′	5′-AAATGACCCCATCGAACACA-3′	5′-/56-FAM/CGCAGACCC/ZEN/CAACAACCAGACT/3IABkFQ/-3′
*Phc3*	5′-GCCTTCATCCATTCTTTGCC-3′	5′-GCTTCATGTTCATTGCACTCAT-3′	5′-/56-FAM/CCCTGAACT/ZEN/CATCTGCAACGTCCTG/3IABkFQ/-3′
*Pknox1*	5′-CTGTTCTTAGATGTTTGCTCGTC-3′	5′-GTCCACTTCAGACATCAGATCA-3′	5′-/56-FAM/TGCACGGCT/ZEN/CTGTTCTTCCAGG/3IABkFQ/-3′
*Rasl11a*	5′-CGACTACGAACCCAACACAG-3′	5′-AGAGAATCCACCATTTGACTGAG-3′	5′-/56-FAM/ATAGCTGGT/ZEN/CCCCCTCCACATAGA/3IABkFQ/-3′
*Rps21*	5′-GAACGTGGCCGAGGTTG-3′	5′-AGACAATTCCATCAGCCTTAGC-3′	5′-/56-FAM/TCATCTGAC/ZEN/TCGCCCATCCTGC/3IABkFQ/-3′
*Rps27*	5′-AGTTCTCCTCGCTCGCA-3′	5′-CCGTGGTGATTTATAGCATCC-3′	5′-/56-FAM/CCAGGCGCT/ZEN/TTTTCTTGTGTTTCCT/3IABkFQ/-3′
*Sdf2l1*	5’-TCGCCGCTATCCAACAAC-3’	5’-CCCAGAACATCGGACTGTC-3’	5’-/56-FAM/TCATCACCC/ZEN/TCACCGTCTTCCC/3IABkFQ/-3’
*Sod3*	5’-GGCAACTCAGAGGCTCTTC-3’	5’-GTAGCAAGCCGTAGAACAAGA-3’	5’-/56-FAM/TTTCCCTCT/ZEN/GGTGAAGTTCAGGCC/3IABkFQ/-3’
*Trem2*	5’-GACCTCTCCACCAGTTTCTC-3’	5’-GCTTCAAGGCGTCATAAGTACA-3’	5’-/56-FAM/TCCCAAGCC/ZEN/CTCAACACCACG/3IABkFQ/-3’
*Trpc6*	5’-CTGGCTCTCATATACTGGTGTG-3’	5’-TCAGCTGCATTCATGACGAG-3’	5’-/56-FAM/AGGAGGCTG/ZEN/CGTGTGCTACAAA/3IABkFQ/-3’
*Vstm2l*	5’-ACTACCTGGCACTTTTCCTG-3’	5’-ATCTCTACATCCTCGCCTGT-3’	5’-/56-FAM/CCGGACACG/ZEN/CACTCTTCACAGA/3IABkFQ/-3’
*Gapdh*	5’-AATGGTGAAGGTCGGTGTG-3’	5’-GTGGAGTCATACTGGAACATGTAG-3’	5’-/56-FAM/TGCAAATGG/ZEN/CAGCCCTGGTG/3IABkFQ/-3’

## Data Availability

Raw data were generated at Showa Medical University. Derived data supporting the findings of this study are available from the corresponding author Y.U. upon reasonable request.

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
