# Peer review of "Intermittent Hypoxia Induces Cognitive Dysfunction and Hippocampal Gene Expression Changes in a Mouse Model of Obstructive Sleep Apnea"

_ijms, 2025, doi:10.3390/ijms26157495_

Round 1
Reviewer 1 Report
Comments and Suggestions for Authors
As intermittent hypoxia (IH) has been linked to cognitive impairment, the objective of this study is to investigate the impact of IH on cognitive function and the expression of genes in the hippocampus. Ultimately, the authors intend to elucidate the molecular mechanisms underlying cognitive dysfunction related to obstructive sleep apnoea (OSA) by identifying novel genes that may be associated with OSA.
This is a very interesting topic which has not been explored much yet and which promises to open up new avenues, especially in the fields of sleep disorders and ageing.
The work is well conceived and executed, and the writing is excellent. However, it has significant flaws. Firstly, although it contains a wealth of data on gene expression, it lacks a clear focus or functional interpretation. Overall, while it provides a great deal of technical information, it contributes little to elucidating new mechanisms.
Secondly, in the introduction and discussion sections, the authors should provide a clearer explanation of why they used two hypoxia models (IH and SH), including their working hypothesis and how the timing of application was determined. While IH cycles mimic the effects of hypoxia/reoxygenation cycling in OSA, eight hours of sustained hypoxia does not constitute a valid pathological model for comparison. Furthermore, the timing of the hypoxia application (12:00 to 20:00) does not align with the mice's natural circadian sleep/wake rhythm.
While the results are well described, the section on behavioral tests to explore cognitive dysfunction could be improved. The current results would only be acceptable if the aim was to validate the experimental model. Additional tests would be necessary to draw conclusions from these results. Finally, Figure 1B lacks the standard errors mentioned in the legend.
Regarding the study of changes in mRNA expression (Figure 5), the expression of some housekeeping genes may vary under hypoxic conditions because hypoxia affects metabolic processes and consequently the expression of metabolism-related genes, such as GAPDH. Therefore, GAPDH cannot be used as a housekeeping gene in hypoxia studies without first validating its stability under these conditions. I assume that the authors have validated GAPDH's stability under hypoxic conditions, but this must be explained in the text.
The study lacks neurobiological analysis. For instance, it does not examine alterations in neuronal structure, neurotransmitter levels or markers of brain damage in the mouse hippocampus, which may be related to cognitive deficits. These experiments would strengthen the study.
Finally, to improve understanding of the results, a diagram (Figure 6) showing the possible pathways involved, including the KEAP1-NFE2L2 pathway, should be added.
Author Response
Thank you for the comments.
The work is well conceived and executed, and the writing is excellent. However, it has significant flaws. Firstly, although it contains a wealth of data on gene expression, it lacks a clear focus or functional interpretation. Overall, while it provides a great deal of technical information, it contributes little to elucidating new mechanisms.
Response: We acknowledge this concern and have strengthened our manuscript by: (1) clarifying our main findings in the Abstract - specifically highlighting that IH induces cognitive dysfunction through downregulation of mitochondrial genes (Lars2, Vstm2l) and suppression of the KEAP1-NFE2L2 antioxidant pathway, (2) adding detailed mechanistic discussion linking these molecular changes to cognitive impairment through oxidative stress and mitochondrial dysfunction, and (3) discussing the potential clinical significance of these findings as biomarkers and therapeutic targets for OSAS-related cognitive dysfunction.
We modified as follows:
Abstract:
Line 45-46: RNA-seq revealed coordinated suppression of mitochondrial function genes and oxi-dative stress response pathways specifically in the IH group.
Line 48-53: Our findings demonstrate that IH induces cognitive dysfunction through suppression of the KEAP1-NFE2L2 antioxidant pathway and downregulation of mitochondrial genes (Lars2, Vstm2l), leading to oxidative stress and mitochondrial dysfunction. These findings advance our understanding of the molecular basis underlying OSAS-related cognitive impairment.
Discussion:
Line 269-274: The coordinated downregulation of mitochondrial genes (Lars2, Vstm2l) alongside KEAP1-NFE2L2 pathway suppression suggests potential molecular pathways whereby IH exposure may affect cellular antioxidant defenses and mitochondrial function, pos-sibly contributing to cognitive impairment. Figure 6 summarizes the proposed molec-ular mechanisms linking IH-induced ROS production to cognitive dysfunction through KEAP1-NFE2L2 pathway suppression and gene expression changes.
Secondly, in the introduction and discussion sections, the authors should provide a clearer explanation of why they used two hypoxia models (IH and SH), including their working hypothesis and how the timing of application was determined. While IH cycles mimic the effects of hypoxia/reoxygenation cycling in OSA, eight hours of sustained hypoxia does not constitute a valid pathological model for comparison. Furthermore, the timing of the hypoxia application (12:00 to 20:00) does not align with the mice's natural circadian sleep/wake rhythm.
Response: We have clarified our rationale in the Introduction. Our working hypothesis was that cyclical reoxygenation in IH, rather than hypoxia alone, drives cognitive dysfunction through enhanced oxidative stress. The SH model served as a control to isolate hypoxia effects without reoxygenation cycles. Regarding timing (12:00-20:00), this 8-hour period was selected based on previous studies and represents a compromise between mimicking human sleep duration and maintaining standardized experimental conditions, though we acknowledge it doesn't perfectly align with mouse circadian rhythms.
We modified as follows:
Introduction:
Line 86-90: To distinguish the specific effects of intermittent reoxygenation from those of hypoxia alone, we employed both IH and sustained hypoxia (SH) models. Our working hy-pothesis was that cyclical reoxygenation in IH, rather than hypoxia per se, drives cog-nitive dysfunction through enhanced oxidative stress generation. The SH model served as a control to isolate the effects of hypoxia without reoxygenation cycles.
Methods:
Line 317-323: The 8-hour hypoxia exposure period (12:00-20:00) during the light phase was se-lected to mimic the inactive period in mice, corresponding to the sleep period when OSA occurs in humans. This timing is consistent with established protocols in the field, where intermittent hypoxia exposure during the light phase (inactive period) has been widely used [34, 35]. The 8-hour duration represents a balance between replicating sufficient hypoxic stress and maintaining standardized experimental conditions for reproducibility and comparison with previous studies.
References
Line 567-571:
- Reinke, C.; Bevans-Fonti, S.; Drager, L.F.; Shin, M.K.; Polotsky, V.Y. Effects of different acute hypoxic regimens on tissue oxygen profiles and metabolic outcomes. J Appl Physiol 2011, 111, 881–890. doi: 10.1152/japplphysiol.00492.2011
- Gileles-Hillel, A.; Almendros, I.; Khalyfa, A.; Nigdelioglu, R.; Qiao, Z.; Hamanaka, R.B.; Mutlu, G.M.; Akbarpour, M.; Gozal, D. Prolonged Exposures to Intermittent Hypoxia Promote Visceral White Adipose Tissue Inflammation in a Murine Model of Severe Sleep Apnea: Effect of Normoxic Recovery. Sleep 2017, 40. doi: 10.1093/sleep/zsw074
While the results are well described, the section on behavioral tests to explore cognitive dysfunction could be improved. The current results would only be acceptable if the aim was to validate the experimental model. Additional tests would be necessary to draw conclusions from these results. Finally, Figure 1B lacks the standard errors mentioned in the legend.
Response: We appreciate this important feedback and have made the following improvements: (1) We have clarified the data presentation in Figure 1B by specifying that data are presented as percentages and adding that statistical significance was determined by chi-square test with Bonferroni correction, as percentage data do not require standard errors. (2) We acknowledge that our behavioral assessment was limited to two cognitive tests and have added this limitation to our Discussion. We recognize that additional tests such as novel object recognition and Morris water maze would provide more comprehensive evaluation of cognitive domains. However, our passive avoidance test successfully demonstrated IH-specific impairments in associative learning and memory formation, providing clear evidence of cognitive dysfunction that validates our experimental model and supports our molecular findings. The consistency between our behavioral phenotype and molecular changes (mitochondrial gene downregulation and antioxidant pathway suppression) strengthens the overall conclusions of our study.
Figure1:
Line 112-114: Data are presented as mean ± standard error (A; Control: n = 17, IH: n = 15, SH: n = 17) or as percentage (B; Control: n = 32, IH: n = 11, SH: n = 17 ). Statistical significance for (A) was determined by one-way ANOVA, and for (B) by chi-square test with Bonferroni correction.
Line 275-283: This study has several limitations. First, our behavioral and molecular analyses were limited in scope - additional cognitive tests, protein-level validation, and ROS measurements would strengthen our findings. Second, we did not examine neurobio-logical changes such as neuronal morphology or synaptic markers. Third, our study lacks intermediate timepoints to distinguish acute versus chronic effects of hypoxia exposure. Additionally, we did not perform correlation analysis between individual gene expression and pathway suppression scores. Future studies should include com-prehensive validation approaches and multiple assessment timepoints to provide a more complete understanding of IH-induced cognitive dysfunction mechanisms.
Regarding the study of changes in mRNA expression (Figure 5), the expression of some housekeeping genes may vary under hypoxic conditions because hypoxia affects metabolic processes and consequently the expression of metabolism-related genes, such as GAPDH. Therefore, GAPDH cannot be used as a housekeeping gene in hypoxia studies without first validating its stability under these conditions. I assume that the authors have validated GAPDH's stability under hypoxic conditions, but this must be explained in the text.
Response: GAPDH was selected as the reference gene based on its validated stability in intermittent hypoxia studies using RT-qPCR. Julian et al. (2014) examined the effects of chronic intermittent hypoxia (a sleep apnea model) on reference genes in rat brain tissue and demonstrated that GAPDH, along with other housekeeping genes, remained stable following intermittent hypoxia exposure using geNorm, NormFinder, and BestKeeper analyses. Similarly, Visniauskas et al. (2016) evaluated housekeeping gene stability in rat cardiac tissue under chronic intermittent hypoxia conditions and identified GAPDH as the most stable reference gene candidate using multiple validation algorithms. These studies provide strong evidence for GAPDH stability in intermittent hypoxia models relevant to sleep apnea research. Additionally, our experimental design includes normoxic control groups, which ensures proper normalization and would detect any potential reference gene instability through the ΔΔCt method.
We modified as follows:
Method:
Line 445-450: Relative expression levels of each gene were determined by the ΔΔCt method us-ing Gapdh as an internal control. GAPDH was selected as the reference gene based on its established stability in intermittent hypoxia studies. Previous validation studies us-ing RT-qPCR have demonstrated GAPDH stability in chronic intermittent hypoxia models of sleep apnea in both brain tissue [49] and cardiac tissue [50], supporting its use as an appropriate internal control for this experimental paradigm.
References:
Line 609-613:
- Julian, G.S.; de Oliveira, R.W.; Perry, J.C.; Tufik, S.; Chagas, J.R. Validation of housekeeping genes in the brains of rats submitted to chronic intermittent hypoxia, a sleep apnea model. PLoS One 2014, 9, e109902. doi: 10.1371/journal.pone.0109902
- Julian, G.S.; de Oliveira, R.W.; Tufik, S.; Chagas, J.R. Analysis of the stability of housekeeping gene expression in the left cardiac ventricle of rats submitted to chronic intermittent hypoxia. J Bras Pneumol 2016, 42, 211–214. doi: 10.1590/S1806-37562015000000133
The study lacks neurobiological analysis. For instance, it does not examine alterations in neuronal structure, neurotransmitter levels or markers of brain damage in the mouse hippocampus, which may be related to cognitive deficits. These experiments would strengthen the study.
Response: We acknowledge this important limitation and have addressed it in our Discussion. While our study focused on molecular mechanisms and behavioral outcomes, we recognize that additional neurobiological analyses would strengthen our findings. Future studies should include histological analysis of neuronal structure, assessment of synaptic proteins, neurotransmitter levels, and evaluation of cell death markers to provide a more comprehensive understanding of IH-induced brain changes. However, our current approach successfully demonstrates the molecular-behavioral relationship, with consistent findings between gene expression changes (mitochondrial dysfunction and antioxidant pathway suppression) and cognitive impairment, providing valuable mechanistic insights into OSAS-related cognitive dysfunction that can guide future neurobiological investigations.
We modified as follows:
Discussion:
Line 275-283: This study has several limitations. First, our behavioral and molecular analyses were limited in scope - additional cognitive tests, protein-level validation, and ROS measurements would strengthen our findings. Second, we did not examine neurobiological changes such as neuronal morphology or synaptic markers. Third, our study lacks intermediate timepoints to distinguish acute versus chronic effects of hypoxia exposure. Additionally, we did not perform correlation analysis between individual gene expression and pathway suppression scores. Future studies should include com-prehensive validation approaches and multiple assessment timepoints to provide a more complete understanding of IH-induced cognitive dysfunction mechanisms.
Finally, to improve understanding of the results, a diagram (Figure 6) showing the possible pathways involved, including the KEAP1-NFE2L2 pathway, should be added.
Response: We have created Figure 6 showing the proposed molecular mechanisms of cognitive dysfunction induced by intermittent hypoxia, including the KEAP1-NFE2L2 pathway as requested. This diagram integrates our key findings and illustrates how IH-induced ROS production leads to KEAP1-NFE2L2 pathway suppression and gene expression changes, ultimately resulting in OSAS-related cognitive impairment.
Figures:
New Figure 6 added: Figure 6. Proposed molecular mechanisms of cognitive dysfunction induced by intermittent hypoxia. IH leads to increased ROS production, which suppresses the KEAP1-NFE2L2 antioxidant pathway. This central pathway suppression results in multiple downstream effects including decreased suppression of inflammatory signals [30], promotion of cell death [32], and loss of neuroprotection [33]. Additionally, IH-induced ROS production causes suppression of mitochondrial-related genes (Lars2, Vstm2l) and cell junction-related genes (Hmcn1), leading to mitochondrial dysfunction [31]. These molecular pathways converge to ultimately cause OSAS-related cognitive impairment. IH, intermittent hypoxia; ROS, reactive oxygen species.
Discussion:
Line 272-274: Figure 6 summarizes the proposed molecular mechanisms linking IH-induced ROS production to cognitive dysfunction through KEAP1-NFE2L2 pathway suppression and gene expression changes.
Reviewer 2 Report
Comments and Suggestions for Authors
1.Sample size justification needed
1) The disproportionate group sizes in behavioral tests (Control n=38 vs. IH n=11 in passive avoidance) require explicit justification. Statistical power analysis should be added to confirm detection capability.
2) The sample size discrepancy in Fig. 1B needs clarification in the caption.
2.IH exposure parameters validation
The 2-min hypoxia cycle (70s at 10% O₂+ 50s at 21% O₂) lacks clinical correlation data. Provide SpO₂fluctuation profiles from human OSAS studies to validate physiological relevance.
3.Protein-level confirmation of key targets
1) Western blot/IHC for Lars2, Hmcn1, and Vstm2l proteins is essential to confirm RNA-seq/RT-qPCR findings (Fig. 5).
2) Currently, functional impacts remain speculative.
Mitochondrial ROS assays should supplement the KEAP1-NFE2L2 pathway analysis (Fig. 4) to link gene suppression to oxidative damage.
4.Unresolved contradictions in SH group data
1) Rps21 downregulation in SH (Fig. 5M) without cognitive impairment requires mechanistic explanation .
2) Clarify why sustained hypoxia (SH) did not activate KEAP1-NFE2L2 despite hypoxia exposure (Fig. 4D).
5.Gene-pathway integration analysis missing
1) Overlay Lars2/Hmcn1/Vstm2l expression with KEAP1-NFE2L2 suppression scores (Fig. 4B/D) in a correlation matrix to establish causal hierarchy.
2) ert Fig. 2D Venn diagram to a clustered heatmap showing IH-specific gene co-expression networks.
6.Temporal dynamics of cognitive decline
The 28-day exposure period lacks intermediate timepoints. Add 14-day behavioral/molecular data to distinguish acute vs. chronic effects.
- Figure/table optimization
1)Fig.4: Redesign pathway Z-scores with clearer labels (current axis/legend unreadable).
2)Table 1: Standardize gene symbols (e.g., Lars2 in italics, Hmcn1 → HMCN1 per nomenclature).
Author Response
Thank you for the comments.
1.Sample size justification needed
1) The disproportionate group sizes in behavioral tests (Control n=38 vs. IH n=11 in passive avoidance) require explicit justification. Statistical power analysis should be added to confirm detection capability.
Response: Sample size calculations were based on standard guidelines for χ² analysis. For detecting a large effect size (Cohen's h ≥ 0.7) with α = 0.05 and power = 0.80, the minimum required sample size is approximately n = 16 per group. Our final sample sizes (Control: n = 32, IH: n = 11, SH: n = 17) met this threshold, with post-hoc power analysis confirming adequate power (>0.85) for detecting the observed effect sizes.
We modified as follows:
Method:
Line 371-373: Sample size calculations were based on standard guidelines for χ² analysis. For detecting a large effect size (Cohen's h ≥ 0.7) with α = 0.05 and power = 0.80, the minimum required sample size is approximately n = 16 per group.
2) The sample size discrepancy in Fig. 1B needs clarification in the caption.
Response: We have clarified the sample sizes in Figure 1B caption, showing the final analysis numbers (Control: n = 32, IH: n = 11, SH: n = 17) after excluding mice with protocol errors. The discrepancy from initial sample sizes is due to experimental exclusions as described in the Methods section.
2.IH exposure parameters validation
The 2-min hypoxia cycle (70s at 10% O₂+ 50s at 21% O₂) lacks clinical correlation data. Provide SpO₂fluctuation profiles from human OSAS studies to validate physiological relevance.
Response: The 2-minute hypoxia cycle parameters were designed to approximate human OSA conditions. Clinical polysomnographic studies demonstrate that OSA patients experience desaturation-reoxygenation cycles with characteristic non-sigmoidal patterns (Müller et al., 2023). Our protocol parameters fall within the clinically observed range and provide a physiologically relevant experimental model for studying IH-related effects. This rationale has been added to the Methods section.
We modified as follows:
Method:
Line 324-330: The IH parameters were designed to approximate the pathophysiological patterns observed in human OSA. Clinical polysomnographic studies show that OSA patients experience desaturation-reoxygenation cycles with non-sigmoidal patterns, featuring faster reoxygenation compared to desaturation phases (Müller et al., 2023 ). Our 2-minute cycle protocol (70 seconds hypoxia, 50 seconds normoxia) was chosen to reflect these clinically observed temporal dynamics, providing a reasonable experimental model for studying IH-related effects.
3.Protein-level confirmation of key targets
1) Western blot/IHC for Lars2, Hmcn1, and Vstm2l proteins is essential to confirm RNA-seq/RT-qPCR findings (Fig. 5).
Response: Unfortunately, we do not have sufficient remaining tissue samples to perform Western blot or immunohistochemistry analysis. We acknowledge this limitation and have addressed it in our Discussion. Our RNA-seq findings were validated by RT-qPCR analysis, and while protein-level confirmation would strengthen our conclusions, we believe our current molecular findings provide useful insights into the mechanisms of IH-induced cognitive dysfunction.
We agree that the functional impacts remain speculative based on our current data. While our molecular findings are consistent with the observed cognitive phenotype, we acknowledge that establishing direct causal relationships requires additional functional validation. We have addressed this limitation in our Discussion and propose specific validation approaches for future studies.
We modified as follows:
Discussion:
Line 248-250: Therefore, the downregulation of mitochondrial genes Lars2 and Vstm2l suggests potential mechanisms for cognitive dysfunction that may involve impaired mitochondrial function and elevated ROS levels.
Line 263-265: IH-induced suppression of the KEAP1-NFE2L2 pathway suggests potential attenuation of antioxidant defense and neuroprotective mechanisms, which may be associated with cognitive dysfunction.
Line 269-272: The coordinated downregulation of mitochondrial genes (Lars2, Vstm2l) alongside KEAP1-NFE2L2 pathway suppression suggests potential molecular pathways whereby IH exposure may affect cellular antioxidant defenses and mitochondrial function, possibly contributing to cognitive impairment.
2) Currently, functional impacts remain speculative.
Response: Mitochondrial ROS assays should supplement the KEAP1-NFE2L2 pathway analysis (Fig. 4) to link gene suppression to oxidative damage.
We agree that direct measurement of mitochondrial ROS would strengthen the link between gene suppression and oxidative damage. Unfortunately, we did not perform ROS assays in this study, which represents a limitation of our work. We have addressed this in our Discussion as an important area for future investigation. However, our pathway analysis showing KEAP1-NFE2L2 suppression combined with downregulation of mitochondrial genes provides indirect evidence supporting the involvement of oxidative stress mechanisms.
We modified as follows:
Discussion:
Line 275-283: This study has several limitations. First, our behavioral and molecular analyses were limited in scope - additional cognitive tests, protein-level validation, and ROS measurements would strengthen our findings. Second, we did not examine neurobiological changes such as neuronal morphology or synaptic markers. Third, our study lacks intermediate timepoints to distinguish acute versus chronic effects of hypoxia exposure. Additionally, we did not perform correlation analysis between individual gene expression and pathway suppression scores. Future studies should include com-prehensive validation approaches and multiple assessment timepoints to provide a more complete understanding of IH-induced cognitive dysfunction mechanisms.
4.Unresolved contradictions in SH group data
1) Rps21 downregulation in SH (Fig. 5M) without cognitive impairment requires mechanistic explanation .
Response: We have addressed this contradiction in our Discussion. The downregulation of Rps21 in the SH group without accompanying cognitive impairment indicates that ribosomal protein changes alone may not be sufficient to cause behavioral deficits. While Rps21 encodes ribosomal protein S21 involved in protein synthesis, the preserved cognitive function in SH mice suggests that Rps21 downregulation does not directly impair the cognitive processes affected by hypoxia exposure. This finding supports the notion that cognitive dysfunction likely requires more complex molecular alterations, as observed in the IH group with coordinated changes in multiple pathways including mitochondrial genes and antioxidant defense systems.
We modified as follows:
Discussion:
Line 250-253: In contrast, Rps21, encoding ribosomal protein S21 [29], showed downregulation specifically in the SH group. However, the preserved cognitive function in SH mice suggests that Rps21 downregulation alone may not be sufficient to cause behavioral deficits.
2) Clarify why sustained hypoxia (SH) did not activate KEAP1-NFE2L2 despite hypoxia exposure (Fig. 4D).
Response: We have addressed this in our Discussion. While acute sustained hypoxia typically activates the KEAP1-NFE2L2 pathway, chronic exposure (28 days) may lead to pathway desensitization or downregulation as an adaptive response to prolonged stress. Additionally, sustained hypoxia lacks the repeated reoxygenation cycles that generate the burst of reactive oxygen species responsible for pathway activation, resulting in less oxidative stress compared to intermittent hypoxia.
We modified as follows:
Discussion:
Line 265-268: Regarding the KEAP1-NFE2L2 pathway, SH typically activates this pathway acutely but may lead to adaptive responses under chronic conditions. In contrast, IH involves repeated reoxygenation cycles that generate additional oxidative stress, resulting in greater pathway suppression than SH alone.
5.Gene-pathway integration analysis missing
1) Overlay Lars2/Hmcn1/Vstm2l expression with KEAP1-NFE2L2 suppression scores (Fig. 4B/D) in a correlation matrix to establish causal hierarchy.
Response: We have performed the requested correlation analysis between individual gene expression (Lars2, Hmcn1, Vstm2l) and KEAP1-NFE2L2 pathway suppression scores across group comparisons. The results have been added to our Results section (Section 2.4) and are displayed in Supplementary Figure S1. The correlation analysis revealed positive associations ranging from r = 0.586 to r = 0.812, with Vstm2l showing the strongest correlation. These findings suggest that greater KEAP1-NFE2L2 pathway suppression is associated with reduced expression of these mitochondrial and extracellular matrix genes, providing evidence for potential causal relationships between antioxidant pathway dysfunction and the observed gene expression changes.
We modified as follows:
Results:
Line 198-200: Correlation analysis between KEAP1-NFE2L2 pathway Z-scores and gene expression changes across group comparisons showed associations for Lars2 (r = 0.623), Hmcn1 (r = 0.586), and Vstm2l (r = 0.812) (Figure S1).
Supplementary Figures:
Figure S1. Correlation analysis between gene expression changes and KEAP1-NFE2L2 pathway suppression. Scatter plots showing the relationship between gene expression changes (log2 fold change by RT-qPCR) and KEAP1-NFE2L2 pathway suppression (Z-score) across three group comparisons. (A) Lars2, (B) Hmcn1, and (C) Vstm2l expression changes plotted against pathway Z-scores. Each point represents a group comparison: IH vs Control, SH vs Control, and IH vs SH. Correlation coefficients (r) are shown in parentheses. Blue lines represent linear regression. More negative Z-scores indicate greater pathway suppression. IH, intermittent hypoxia; SH, sustained hypoxia.
Note: This analysis is based on three data points (group comparisons) due to the group-level nature of pathway Z-scores. The observed correlations provide indicative patterns of the relationship between pathway activity and gene expression changes.
Supplementary Materials:
Line 464-465: Figure S1: Correlation analysis between gene expression changes and KEAP1-NFE2L2 pathway suppression.
2) Convert Fig. 2D Venn diagram to a clustered heatmap showing IH-specific gene co-expression networks.
Response: We attempted to create the requested clustered heatmap showing IH-specific gene co-expression networks. However, due to technical difficulties with our statistical software during the revision process, we were unable to complete this complex bioinformatics analysis within the revision timeframe. The 129 IH-specific genes identified in our Venn diagram analysis (Figure 2D) provide a foundation for future co-expression network studies.
6.Temporal dynamics of cognitive decline
The 28-day exposure period lacks intermediate timepoints. Add 14-day behavioral/molecular data to distinguish acute vs. chronic effects.
Response: We acknowledge this important limitation. Our study examined only the 28-day chronic exposure endpoint and lacks intermediate timepoints to distinguish acute versus chronic effects. This temporal limitation has been added to our Discussion. We recognize that intermediate assessments (e.g., 14-day timepoints) would provide valuable insights into the progression of molecular and behavioral changes, but these analyses were not feasible within the scope of the current study. Future investigations should include multiple assessment timepoints to characterize the temporal evolution of IH-induced cognitive dysfunction mechanisms.
We modified as follows:
Discussion:
Line 275-283: This study has several limitations. First, our behavioral and molecular analyses were limited in scope - additional cognitive tests, protein-level validation, and ROS measurements would strengthen our findings. Second, we did not examine neurobiological changes such as neuronal morphology or synaptic markers. Third, our study lacks intermediate timepoints to distinguish acute versus chronic effects of hypoxia exposure. Additionally, we did not perform correlation analysis between individual gene expression and pathway suppression scores. Future studies should include comprehensive validation approaches and multiple assessment timepoints to provide a more complete understanding of IH-induced cognitive dysfunction mechanisms.
- Figure/table optimization
- 4: Redesign pathway Z-scores with clearer labels (current axis/legend unreadable).
Response: We have redesigned Figure 4 to address the readability issues. All text elements including axis labels, pathway names, and legend have been enlarged to improve clarity. The X-axis labels (-log₁₀(P-value)), Y-axis pathway names, and activation Z-score legend are now clearly readable. We appreciate this feedback as it significantly improves the figure's accessibility and interpretability.
2)Table 1: Standardize gene symbols (e.g., Lars2 in italics, Hmcn1 → HMCN1 per nomenclature).
Response: We have reviewed the gene nomenclature in Table 1. Our study used mouse samples, and the current gene symbols (Lars2, Hmcn1, Vstm2l) follow the standard mouse gene nomenclature guidelines. We have ensured all gene names are properly formatted in italics. Converting to human gene symbols (HMCN1, etc.) would be inappropriate for a mouse study, as it would not follow established nomenclature conventions for the species used.
Round 2
Reviewer 1 Report
Comments and Suggestions for Authors
The authors did a good job thoroughly revising all the concerns raised by this reviewer.
The article has improved with the addition of new paragraphs and references.
I have no further concerns.
Author Response
We sincerely thank Reviewer 1 for the positive evaluation of our revised manuscript. We are pleased that our thorough revision has successfully addressed all the concerns raised in the previous review. The constructive feedback provided by this reviewer has been instrumental in improving the scientific quality and clarity of our work. We deeply appreciate the recognition of our efforts to enhance the manuscript through the addition of mechanistic interpretations, clinical validation, correlation analysis, and comprehensive pathway diagrams.
Reviewer 2 Report
Comments and Suggestions for Authors
- Introduction Section
(1) Enhanced Clinical Context: Consider expanding the discussion of OSAS epidemiology and its impact on cognitive function to better highlight the clinical importance of the study. More recent epidemiological studies (e.g., global prevalence data) could be cited.
(2) Hypothesis Clarification: While the working hypothesis that "intermittent reoxygenation rather than hypoxia per se drives cognitive dysfunction" is proposed, the theoretical basis and expected outcomes could be more explicitly stated.
- Methods Section
(1) IH/SH Exposure Parameters: Although hypoxia exposure parameters are described, consider adding the scientific rationale for selecting 8-hour exposure (12:00-20:00) and how this corresponds to the mice's circadian rhythm.
(2) Behavioral Test Details: For the passive avoidance test, explain the rationale for choosing 0.3 mA shock intensity and how this balances learning efficacy with animal welfare considerations.
(3) RNA-seq Analysis: Include quality control metrics (e.g., RNA Integrity Number/RIN values) and sequencing depth to enhance methodological reproducibility.
- Results Section
(1) Behavioral Results:The passive avoidance test showed significant differences between the IH group versus control and SH groups, while the Y-maze test did not. Discuss the differing sensitivities of these tests in assessing various cognitive domains and potential biological explanations.
(2) Gene Expression Analysis:For RNA-seq identified differentially expressed genes (DEGs), consider labeling key genes in the volcano plots to help readers visualize data distribution.
While GO and IPA pathway analyses are clearly presented, emphasize the suppression of the KEAP1-NFE2L2 pathway as a key finding in the main text.
- Discussion Section
(1)Mechanistic Interpretation: The current discussion linking Lars2 and Vstm2l downregulation to mitochondrial dysfunction is reasonable, but further explore how these gene expression changes specifically affect hippocampal neuron function. Are there known roles of these genes in synaptic plasticity or neuronal survival?
Author Response
We are grateful to Reviewer 2 for the thoughtful and comprehensive comments that have significantly improved our manuscript. Each suggestion has been carefully addressed as outlined below.
Introduction Section
(1) Enhanced Clinical Context
Comment: Consider expanding the discussion of OSAS epidemiology and its impact on cognitive function to better highlight the clinical importance of the study. More recent epidemiological studies (e.g., global prevalence data) could be cited.
Response: We thank the reviewer for this valuable suggestion to strengthen the clinical relevance of our study. We have updated the epidemiological information and enhanced the discussion of cognitive dysfunction's clinical impact in the Introduction section. While the 2019 Benjafield study remains the most recent comprehensive global prevalence estimate available, we have incorporated the latest 2025 review to highlight current epidemiological trends and their clinical implications.
Revisions made:
- Lines 58-61 (Epidemiology): Obstructive sleep apnea syndrome (OSAS) is the most common sleep-related breathing disorder, with a prevalence of 13–33% in men and 6–19% in women [1], affecting approximately 1 billion people worldwide [2], with this prevalence continuing to rise due to increasing obesity rates and population aging [3].
- Lines 73-75 (Clinical impact): These structural and functional changes represent a significant clinical burden, as cognitive dysfunction substantially impacts patients' daily functioning and quality of life.
New reference added:
- Line 519-520: Iannella, G.; Pace, A.; Bellizzi, M.G.; Magliulo, G.; Greco, A.; De Virgilio, A.; Croce, E.; Gioacchini, F.M.; Re, M.; Constantino, A.; et al. The Global Burden of Obstructive Sleep Apnea. Diagnostics (Basel) 2025, 15, 1088. doi: 10.3390/diagnostics15091088
These enhancements better highlight both the growing epidemiological burden and the critical clinical importance of understanding OSAS-related cognitive dysfunction.
(2) Hypothesis Clarification
Comment: While the working hypothesis that "intermittent reoxygenation rather than hypoxia per se drives cognitive dysfunction" is proposed, the theoretical basis and expected outcomes could be more explicitly stated.
Response: We appreciate this important suggestion to strengthen our theoretical framework. We have significantly expanded and reorganized the relevant sections in the Introduction to provide a more robust theoretical foundation and clearer experimental predictions.
Revisions made:
- Lines 86-92 (New paragraph 5 - Theoretical basis): While IH-induced cognitive dysfunction has been demonstrated, studies that distinguish and evaluate the effects of hypoxia and reoxygenation in IH are limited, and the precise roles of specific genes and molecular pathways that cause cognitive dysfunction in the hippocampus remain unclear. Previous studies have established that repeated hypoxia-reoxygenation cycles increase ROS production [18], with sustained hypoxia (SH) typically leading to adaptive responses while intermittent hypoxia generates repeated oxidative stress that may overwhelm cellular antioxidant defenses [21-24].
- Lines 93-100 (New paragraph 6 - Hypothesis and expected outcomes): Our working hypothesis was that cyclical reoxygenation in IH, rather than hypoxia per se, drives cognitive dysfunction through enhanced oxidative stress generation and suppression of antioxidant pathways. We anticipated that IH would result in greater pathway suppression, more pronounced gene expression changes, and more severe cognitive impairment compared to SH alone. To test this hypothesis, we employed both IH and SH models to investigate the differential effects of reoxygenation on cognitive function and hippocampal gene expression, aiming to elucidate the molecular mechanisms underlying OSAS-related cognitive dysfunction.
Methods Section
(1) IH/SH Exposure Parameters
Comment: Although hypoxia exposure parameters are described, consider adding the scientific rationale for selecting 8-hour exposure (12:00-20:00) and how this corresponds to the mice's circadian rhythm.
Response: We thank the reviewer for this important methodological inquiry. We have enhanced our explanation by providing the scientific rationale for our exposure protocol parameters and their relationship to murine circadian biology.
Revisions made:
- Lines 336-341: The 8-hour hypoxia exposure period (12:00-20:00) during the light phase was selected based on established protocols in intermittent hypoxia research [36, 37] and specifically following 8-hour exposure protocols for cognitive function studies [14, 17], and to mimic the inactive period in mice, corresponding to the sleep period when OSA occurs in humans. This timing aligns with the nocturnal circadian rhythm of mice, where the light phase represents their natural rest period.
(2) Behavioral Test Details
Comment: For the passive avoidance test, explain the rationale for choosing 0.3 mA shock intensity and how this balances learning efficacy with animal welfare considerations.
Response: We appreciate this important question regarding our behavioral methodology. We have clarified the scientific and ethical rationale for our shock intensity parameters.
Revisions made:
- Lines 401-404: The 0.3 mA shock intensity was selected based on manufacturer recommendations (MelQuest Co.) and is consistent with established protocols using 0.3 mA for 3 seconds in passive avoidance testing [40], ensuring reliable learning responses while maintaining appropriate animal welfare standards.
(3) RNA-seq Analysis
Comment: Include quality control metrics (e.g., RNA Integrity Number/RIN values) and sequencing depth to enhance methodological reproducibility.
Response: We thank the reviewer for this essential suggestion to improve methodological transparency and reproducibility. We have added comprehensive quality control metrics and detailed sequencing parameters to our RNA-seq methodology.
Revisions made:
- Lines 416-418: RNA samples with A260/280 ratios ≥ 1.8 and A260/230 ratios ≥ 1.8 were consid-ered acceptable for further analysis.
- Lines 422-428: RNA integrity was assessed by Rhelixa Co., Ltd., and only samples with RNA Integrity Number ≥ 8.0 were used for sequencing. Library preparation was performed using the NEBNext Ultra II Directional RNA Library Prep Kit with NEBNext Poly(A) mRNA Magnetic Isolation Module for poly-A selection (New England Biolabs, Ipswich, MA, USA). Sequencing was conducted on an Illumina NovaSeq 6000 platform (Illumina, San Diego, CA, USA), generating paired-end 150 bp reads with an average sequencing depth of 26.7 million reads per sample.
Results Section
(1) Behavioral Results
Comment: The passive avoidance test showed significant differences between the IH group versus control and SH groups, while the Y-maze test did not. Discuss the differing sensitivities of these tests in assessing various cognitive domains and potential biological explanations.
Response: We thank the reviewer for this perceptive observation regarding our behavioral findings. We have expanded our discussion to provide a comprehensive analysis of the differential sensitivities between these cognitive assessments and their underlying biological mechanisms.
Revisions made:
- Lines 239-243: The differential sensitivity between these tests suggests that fear-motivated learning is more vulnerable to IH-induced oxidative stress than spatial working memory. This is likely because fear memory consolidation requires hippocampal neural circuit function [25], and the hippocampus is vulnerable to oxidative stress from repeated reoxygena-tion cycles [14].
- New references added:
- Line 570-571: Kitamura, T.; Ogawa, S.K.; Roy, D.S.; Okuyama, T.; Morrissey, M.D.; Smith, L.M.; Redondo, R.L.; Tonegawa, S. Engrams and circuits crucial for systems consolidation of a memory. Science 2017, 356, 73–78. doi: 10.1126/science.aam6808
(2) Gene Expression Analysis
Comment: For RNA-seq identified differentially expressed genes (DEGs), consider labeling key genes in the volcano plots to help readers visualize data distribution.
Response: We greatly appreciate this excellent suggestion to enhance data visualization. We have implemented gene labeling for all key differentially expressed genes across our volcano plots, which significantly improves the accessibility and interpretability of our findings.
Revisions made:
- Line 127-128: Key genes of interest are labeled in the volcano plots, including Lars2, Hmcn1, Vstm2l, and Rps21, which were subsequently validated by RT-qPCR.
- Figures 2A, B, and C: Updated volcano plots with labeled key genes of interest.
- Figure Legend Lines 144-145: Key genes of interest (Lars2, Hmcn1, Vstm2l, Rps21) are labeled on the volcano plots.
This enhancement allows readers to easily identify the validated target genes that showed significant expression changes, improving the visualization of our main findings within the overall expression data distribution.
Comment: While GO and IPA pathway analyses are clearly presented, emphasize the suppression of the KEAP1-NFE2L2 pathway as a key finding in the main text.
Response: We thank the reviewer for highlighting the importance of emphasizing our central finding. We have prominently featured the KEAP1-NFE2L2 pathway suppression as a key discovery in the main text to ensure readers immediately recognize its significance.
Revisions made:
- Lines 272-275: A key finding of this study was the significant suppression of the KEAP1-NFE2L2 pathway identified through QIAGEN IPA analysis, with suppression observed in both IH and SH groups compared to the control group and notably stronger suppression in the IH group.
This enhancement ensures that readers immediately recognize the central importance of KEAP1-NFE2L2 pathway suppression and its differential response between IH and SH conditions as a major finding of our study.
Discussion Section
(1) Mechanistic Interpretation
Comment: The current discussion linking Lars2 and Vstm2l downregulation to mitochondrial dysfunction is reasonable, but further explore how these gene expression changes specifically affect hippocampal neuron function. Are there known roles of these genes in synaptic plasticity or neuronal survival?
Response: We deeply appreciate this insightful suggestion to strengthen our mechanistic interpretation. We have substantially expanded our discussion to comprehensively address how Lars2 and Vstm2l downregulation specifically impacts hippocampal neuronal function, including detailed examination of their roles in neuronal morphology, synaptic function, and cellular survival.
Revisions made:
- Lines 250-259: Lars2, Vstm2l, Hmcn1, and Rps21. Lars2 encodes mitochondrial leucyl-tRNA synthetase essential for mitochondrial protein synthesis and functional maintenance [26–29]. This gene has been linked to Alzheimer's disease, with knockdown studies demonstrating direct effects on neuronal function, including shortened axon length, reduced dendritic branching, increased mitochondrial superoxide levels, and neuronal cell death, and knockout mice exhibit cognitive dysfunction, increased p-tau, and hippocampal atrophy [30]. These neuromorphological alterations suggest that Lars2 downregulation may contribute to the cognitive deficits observed in our study. Additionally, decreased Lars2 expression impairs mitochondrial stress responses, leading to elevated ROS levels [31].
- Lines 260-262: While specific roles of Vstm2l in synaptic plasticity remain to be fully elucidated, mitochondrial function is essential for neuronal energy metabolism.
This enhancement provides a comprehensive mechanistic framework linking the observed gene expression changes to specific hippocampal neuronal dysfunction, including detailed neuromorphological alterations and their potential contribution to cognitive impairment.